# Non-linear photoconductivity of strongly driven graphene

Lukas Broers[1,2⋆] and Ludwig Mathey[1,2,3]

**1** Center for Optical Quantum Technologies, University of Hamburg, Hamburg, Germany
**2** Institute for Quantum Physics, University of Hamburg, Hamburg, Germany
**3** The Hamburg Center for Ultrafast Imaging, Hamburg, Germany

⋆ lbroers@physnet.uni-hamburg.de

## Abstract

We present the non-linear DC photoconductivity of graphene under strong infra-red (IR) radiation. The photoconductivity is obtained as the response to a strong DC electric field, with field strengths outside of the linear-response regime, while the IR radiation is described by a strong AC electric field. The conductivity displays two distinct regimes in which either the DC or the AC field dominates. We explore these regimes and associate them with the dynamics of driven Landau-Zener quenches in the case of a large DC field. In the limit of a large AC field, we describe the conductivity in a Floquet picture and compare the results to the closely related Tien-Gordon effect. We present analytical calculations for the non-linear differential photoconductivity, for both regimes based on the corresponding mechanisms. As part of this discussion of the non-equilibrium state of graphene, we present analytical estimates of the conductivity of undriven graphene as a function of temperature and DC bias field strength that show very good agreement with our simulations.



# 1 Introduction

Graphene displays a wide range of remarkable properties, which have been the subject of basic research and of technological interest, since its discovery [1–6]. Key interests have been the electronic transport [7–29] as well as optical properties [30–37], including the effects of coherent irradiation [38–45]. For instance, circularly polarized light generates topological Floquet band gaps which leads to an anomalous Hall effect [46–49]. Third-order susceptibilities [50–53] capture non-linear optical transport phenomena, such as higher-harmonic generation. Quasi-classical approaches have provided descriptions that explain various electrodynamical responses [54–56]. Further, coherent control of transport has been demonstrated by driving graphene with short pulses, which leads to charge-envelope-phase-dependent Stückelberg interferometrically induced currents [57–60], which relate to Landau-Zener transition processes [61, 62]. The coherent destruction of tunneling [63–65] has been proposed in strongly driven graphene [66], as well as photon-assisted tunneling [67]. Given the recent studies of light-controlled phenomena in graphene, a comprehensive characterization of the non-linear photoconductivity and coherent dynamics and transport in periodically driven graphene is imperative, and is also motivated by future technologies based on high-intensity driving.

In this work, we present the non-linear longitudinal DC photoconductivity of monolayer graphene driven with linearly polarized terahertz radiation at the charge neutrality point. The polarization of the AC driving field is parallel to the DC bias polarization. We utilize a Master equation approach that explicitly models the dissipative properties of the material, see [49]. We identify a rich structure in the differential conductivity which features two limits, in which either the DC bias field strength or the AC driving field strength dominates. These two limits are separated by a regime in which the two fields are comparatively strong, and the dynamics display a subtle competition. In the regime in which the DC bias field dominates, we describe the dynamics via Landau-Zener (LZ) transitions that are modulated by the driving field, and affected by the dissipative properties of the system. Due to the modulation by the driving field, dynamical patterns in momentum space, in the vicinity of the Dirac points, emerge with a periodicity that is equal to the accumulated momentum shift during one driving cycle. We present an analytical solution to leading order in the transverse momentum component, i.e. the momentum component orthogonal to the DC bias direction relative to one of the Dirac points. This analytical solution reproduces the characteristic patterns in the differential photoconductivity. In the absence of the driving field, we estimate this current density pattern analytically and derive an expression for the non-linear conductivity of strongly biased undriven graphene. Further, we provide analytical calculations for the dependence on temperature of the undriven conductivity. For a weak DC bias field, we find that the conductivity scales linearly with temperature down to small temperatures at which the conductivity converges to the analytical minimal conductivity of $\frac{\pi e^2}{2h}$. In the regime in which the driving field dominates, we find that the differential photoconductivity displays a type of checkerboard pattern. The DC bias field provides a shift in momentum space across the strongly driven Floquet band structure, such that the conductivity pattern emerges from an interplay of Floquet and transport dynamics. We note that the checkerboard pattern in the differential conductivity shows strong qualitative similarities to predictions for this system of the distinct Tien-Gordon effect [68].

# 2 Methods

We consider monolayer graphene in the presence of a constant electric field, i.e. a direct (DC) bias, in addition to continuous terahertz radiation, i.e. an alternating (AC) driving field, which

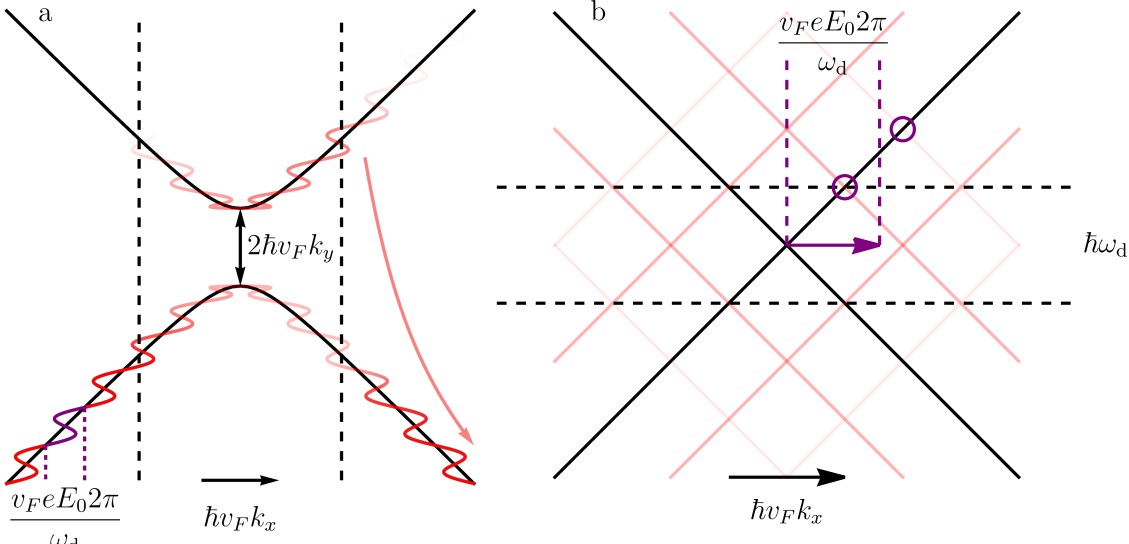

Figure 1: Illustrations of the dynamics in the two regimes of either dominating DC field strength $E_0$ or AC field strength $E_d$. Panel (a) shows the structure of the modified Landau-Zener quench in the limit of dominating DC field strength $E_0$. An electron in the ground state with initial momentum far away from the gap is accelerated due to $E_0$ and displays quenched dynamics across the gap. Due to the additional AC field, the quench is modulated. Panel (b) shows the structure of the Floquet band structure in the limit of large and dominating AC field strength $E_d$. The DC field strength $E_0$ accelerates electrons across this effective band structure.

is linearly polarized in parallel to the DC bias field. We write the linearized Hamiltonian for a single momentum mode $\mathbf{k} = (k_x, k_y)$ near one of the Dirac points, as

$$\frac{1}{\hbar} H_{\mathbf{k}}(t) = v_F \left( k_x + \frac{eE_0}{\hbar} t + \frac{eE_d}{\hbar \omega_d} \cos(\omega_d t) \right) \tau_z \sigma_x + v_F k_y \sigma_y. \tag{1}$$

$\tau_z$ indicates the Dirac point at which this model is valid, taking either the value 1 or $-1$. Here $e$ is the elementary charge, $\hbar$ is the reduced Planck constant and $v_F \approx 10^6 \mathrm{m\,s^{-1}}$ is the Fermi velocity of graphene. $E_0$ is the DC field strength, $E_d$ is the AC field strength and $\omega_d$ is the driving frequency. $\sigma_x$ and $\sigma_y$ are Pauli matrices. We propagate the density operator of the system using the Lindblad-von Neumann master equation

$$\dot{\rho}_{\mathbf{k}} = \frac{i}{\hbar} [\rho_{\mathbf{k}}, H_{\mathbf{k}}(t)] + \sum_l \gamma_l \left( L_l \rho_{\mathbf{k}} L_l^\dagger - \frac{1}{2} \{ L_l^\dagger L_l, \rho_{\mathbf{k}} \} \right). \tag{2}$$

The indices $l$ of the Lindblad operators describe the dissipative processes of spontaneous decay and excitation, dephasing and incoherent exchange to an electronic backgate. The associated dissipation rates are $\gamma_-$, $\gamma_+$, $\gamma_z$ and $\gamma_{bg}$, respectively. The temperature $T$ of the system enters the model through Boltzmann factors of conjugate processes, e.g. $\gamma_+ = \gamma_- \exp\{-\frac{2\epsilon}{k_B T}\}$, where $\epsilon$ is the instantaneous eigenenergy scale of the driven Hamiltonian. The Lindblad operators $L_l$ act in the instantaneous eigenbasis of the Hamiltonian. For further details of this method we refer to App. C and previous works [49, 69, 70]. Throughout this work we use the parameters $\omega_d = 2\pi \times 12\,\mathrm{THz}$, $\gamma_z = 11.25\,\mathrm{THz}$, $\gamma_+ + \gamma_- = 5\,\mathrm{THz}$, $\gamma_{bg} = 12.5\,\mathrm{THz}$ and $T = 80\,\mathrm{K}$ unless stated otherwise. We explore values for the electric field strengths $E_0$ and $E_d$ up to a few megavolts per meter.

We calculate the total longitudinal current of the system by integrating the momentum-resolved contributions of the current operator

$$j_x = e v_F \tau_z \sigma_x, \tag{3}$$

over momentum space. Note that the periodicity of the AC field requires integration over one driving period to obtain the DC component of the total current. Due to the structure of the electric field in Eq. 1 which fully acts along the $x$-direction, the integrated contribution to the total current is equal at the two Dirac points. Therefore, it is

$$J_x = n_v n_s \frac{e v_F \omega_{\mathrm{d}}}{8\pi^3} \sum_{\mathbf{k} \in \mathcal{D}} \int_\tau^{\tau + \frac{2\pi}{\omega_{\mathrm{d}}}} \mathrm{Tr}(\sigma_x \rho_{\mathbf{k}}(t)) dt \, \Delta k_x \Delta k_y, \tag{4}$$

with the valley- and spin-degeneracy $n_v = n_s = 2$. $\Delta k_x$ and $\Delta k_y$ are the numerical discretization of momentum space. $\mathcal{D}$ is a sufficiently large neighborhood in momentum space around the Dirac point $K$ to ensure convergence. $\tau$ is a time that is large enough that the system has formed a steady state in the comoving frame $k_x \to k_x - \frac{e}{\hbar} E_0 t$. Rather than considering finite-order non-linearities, we calculate the derivative of the total current of the system with respect to the Fourier coefficients of the electric field. This gives the differential conductivity

$$G_{xx} = \frac{dJ_x}{dE_0}, \tag{5}$$

and in the absence of driving

$$G_{xx}^0 = G_{xx}|_{E_{\mathrm{d}}=0}. \tag{6}$$

Further, we consider the differential photoconductivity with respect to the AC field strength

$$g_{xx} = \frac{dG_{xx}}{dE_{\mathrm{d}}}. \tag{7}$$

We obtain the derivatives of Eqs. 5 and 7 numerically as central finite differences. Since in this model we are considering graphene at its charge neutrality point, i.e. at vanishing chemical potential, the conductivity we calculate is commonly referred to as the minimal conductivity of graphene, in relation to the possible enhancement of the conductivity by increasing the charge carrier density by means of a non-zero chemical potential.

## 3 Results

We calculate the overall current $J_x$ as a function of the DC field strength $E_0$ and the AC field strength $E_{\mathrm{d}}$ up to values of $4\,\mathrm{MV\,m^{-1}}$ and $20\,\mathrm{MV\,m^{-1}}$, respectively. This range of the DC bias field includes the nonlinear DC conductivity of driven graphene well beyond the linear response regime. In Fig. 2 (a) and (b) we show the differential photoconductivity $g_{xx}$ and the change in differential conductivity

$$\Delta G_{xx} = G_{xx} - G_{xx}^0, \tag{8}$$

respectively. The structure of these observables is rich and there are two distinct regimes which correspond to the cases of $E_{\mathrm{d}} \gg E_0$ and $E_0 \gg E_{\mathrm{d}}$, in which qualitative structural dependencies can clearly be identified. Depending on whether the DC field or the AC field dominates, the steady state dynamics change significantly. As we discuss below, the AC and DC fields are on equal scales when $E_{\mathrm{d}} = 2\pi E_0$. This condition is met when the momentum shift due to the DC

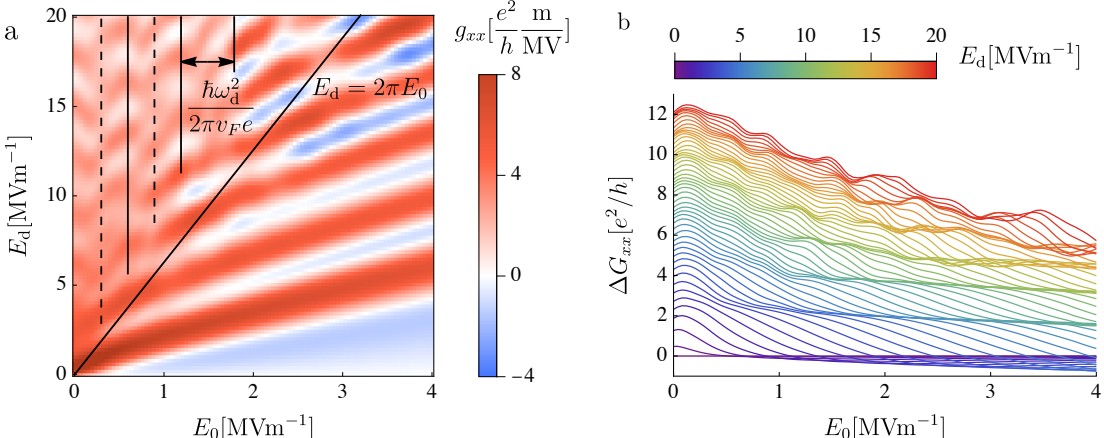

Figure 2: The differential conductivities of strongly driven graphene. Panel (a) shows the differential photoconductivity $g_{xx}$ as a function of the DC field strength $E_0$ and the AC field strength $E_d$. The diagonal line is given by $E_d = 2\pi E_0$ and separates the results into the two regimes where either the DC field strength $E_0$ or the AC field strength $E_d$ dominates. Panel (b) shows the change in differential conductivity $\Delta G_{xx}$ as a function of $E_0$ for various values of $E_d$.

field that is accumulated during one driving period is equal to the momentum displacement amplitude due to the AC field. We indicate this condition with the diagonal line in Fig. 2 (a). The structure of the differential photoconductivity $g_{xx}$ in these two regimes is intricate and can be understood from different perspectives in analyzing the corresponding dynamics. The following subsections discuss these two regimes.

## 3.1 Dominant DC field

We first analyze the regime in which the DC field strength $E_0$ is significantly larger than the AC field strength $E_d$. In this regime, the differential photoconductivity $g_{xx}$ in Fig. 2 (a) shows a striped pattern as a function of $E_d$ and $E_0$ which leads to step-like features in the change in differential conductivity $\Delta G_{xx}$ that we show in Fig. 2 (b). The dynamics of this case are captured naturally in the comoving frame $k_x \rightarrow k_x - \frac{eE_0}{\hbar}t$, produced by the large momentum shift due to the DC field $E_0$. In this frame, an electron with a momentum far to one side of the Dirac point and initially in thermal equilibrium accelerates due to the DC field and eventually passes the Dirac point. These dynamics are a type of Landau-Zener (LZ) quench across the gap given by the transverse momentum component, i.e. $\Delta = 2\hbar v_F k_y$. The driving term is linearly polarized in parallel with the bias field, such that the LZ quench is further modified by an undulating motion in $k_x$ as depicted in Fig. 1 (a). As the relative phase of the AC field during the quench depends on the initial value of $k_x$, the transition probability is periodic in momentum with $eE_0\hbar^{-1}2\pi\omega_d^{-1}$, which is the momentum shift induced by the DC field during one driving period. Therefore, the temporal periodicity of the AC field leads to a periodic current density pattern in momentum space.

In Fig. 3, we show examples of such patterns in the quantity

$$\overline{\Delta j_x}(\mathbf{k}) = \frac{\omega_d}{2\pi} \int_\tau^{\tau + \frac{2\pi}{\omega_d}} \Delta j_x(\mathbf{k}) dt \,, \tag{9}$$

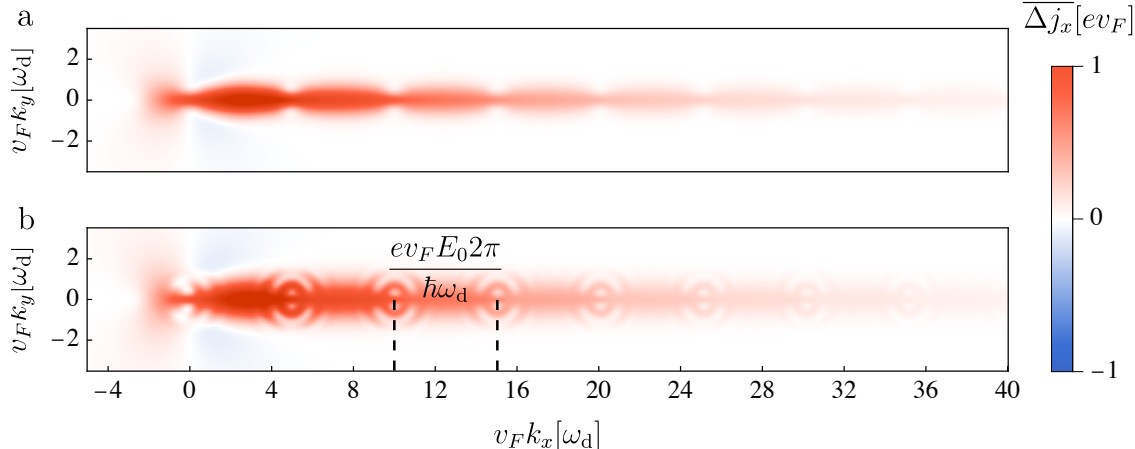

Figure 3: The momentum-resolved photocurrent density in the presence of a strong DC field of $E_0 = 3\,\mathrm{MV\,m^{-1}}$ and an AC field strength of $E_\mathrm{d} = 2\,\mathrm{MV\,m^{-1}}$ (a) and $E_\mathrm{d} = 6\,\mathrm{MV\,m^{-1}}$ (b) averaged over one driving period. Here, all dissipation coefficients are reduced by a factor of five compared to the values in the main text, for visual clarity. The dashed lines indicate the momentum shift that is accumulated during one driving period, which corresponds to the periodic patterns that occurs due to the AC field.

which is the time-average of the momentum-resolved photocurrent density

$$\Delta j_x(\mathbf{k}) = j_x(\mathbf{k}) - j_x(\mathbf{k})|_{E_0,E_\mathrm{d}=0}\,, \tag{10}$$

in which we subtract the equilibrium current density which integrates to zero. $\tau$ is a time large enough that a steady state has formed in the comoving frame $k_x \to k_x - \frac{eE_0}{\hbar}t$. In the case of large values of $E_0$ which leads to displacement in momentum space at a high rate compared to the dissipation, the steady state current density pattern stretches very far across momentum space before significantly decaying. Note that for increasing AC field strength $E_\mathrm{d}$ the pattern becomes more intricate.

   In the picture of modified LZ quenches, the integrated current in Eq. 4 consists of a contribution close to the gap, and a much larger contribution from the decaying tail of the current density pattern as is clearly visible in Fig. 3. We neglect the first part and estimate the current as a product of the periodic current density pattern and exponential decay with a dissipation rate $\Gamma = \frac{1}{2}\gamma_- + \gamma_\mathrm{bg}$. We present the details of this calculation in App. A. We write the total current as

$$J_x^\mathrm{LZ} \approx n_\nu n_s \frac{ev_F\omega_\mathrm{d}}{8\pi^3} \int_0^{\frac{eE_0 2\pi}{\hbar\omega_\mathrm{d}}} \int_{\mathbb{R}^2} 2P(k_{x,0}+\kappa, k_y)e^{-\Gamma\frac{\hbar k_x}{eE_0}}\,dk_x dk_y d\kappa\,, \tag{11}$$

where $P(k_x, k_y)$ is the transition probability into the excited state of a system initially in the ground state at momentum $k_x$. It is

$$P(k_x, k_y) = \lim_{t\to\infty} |\langle +| U(t,0)|-\rangle|^2\,, \tag{12}$$

where $U(t_2, t_1)$ is the time-evolution operator of the Hamiltonian in Eq. 1 from time $t_1$ to $t_2$ and $|\pm\rangle$ are the eigenstates of the $\sigma_x$ Pauli matrix. $k_{x,0}$ is an initial momentum component that is negative and large enough such that the dynamics are initially adiabatic, in order to capture the quenched dynamics in their entirety. $\kappa$ is an additional momentum offset in order to average over the periodicity of the density pattern.

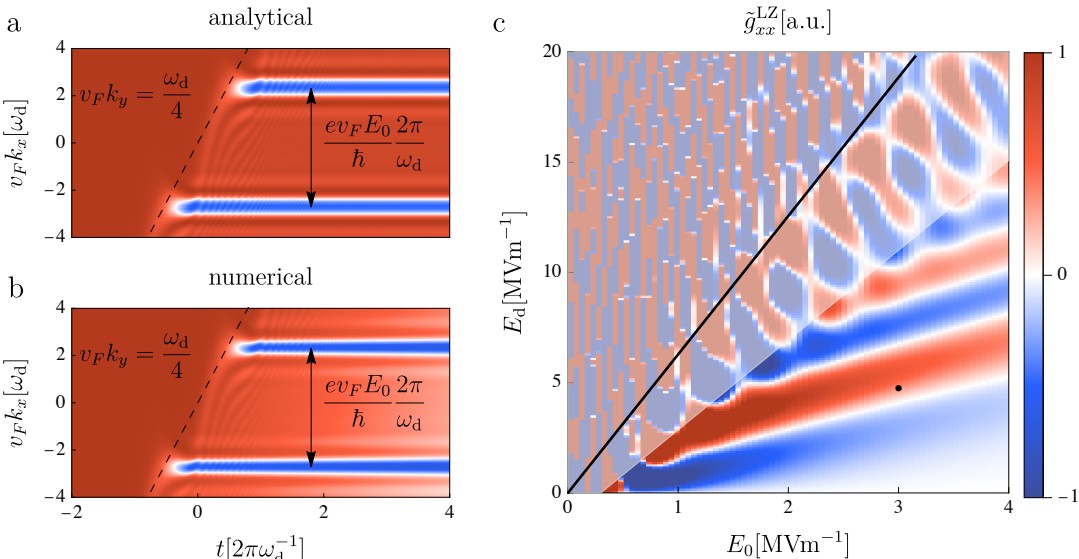

Figure 4: Comparison of the quenched dynamics for $E_0 = 3\,\mathrm{MV\,m^{-1}}$ and $E_\mathrm{d} = 4.75\,\mathrm{MV\,m^{-1}}$ as a function of the initial momentum component $k_x$ at $v_F k_y = \omega_\mathrm{d}/4$. Panel (a) shows the analytical solution to the modified Landau-Zener quenched dynamics (See App. A) that lead to the expression in Eq. 13. Panel (b) shows the numerical results in the presence of weak dissipation. A weak exponential decay due to the dissipation is visible, and the overall structure agrees well with the analytical estimate in panel (a). Panel (c) shows the analytical estimate for the differential photoconductivity from the modified Landau-Zener quench to leading order in $k_y$ as given by Eq. 17. The striped structure agrees qualitatively with the numerical results in Fig. 2 (a) for large values of $E_0$. The black line is given by $E_\mathrm{d} = 2\pi E_0$. The black dot indicates the parameters used in panels (a) and (b). The shaded area indicates the regime in which Eq. 17 is not a valid approximation.

We calculate $P(k_x, k_y)$ from the modified LZ problem (See App. A). To leading order in $k_y$ it is

$$|\langle +|U(t,0)|-\rangle|^2 = \exp\left\{-\frac{\pi\hbar v_F k_y^2}{eE_0}\left|\int_0^t e^{i\frac{2}{\hbar}\Pi(t')}dt'\right|^2\right\},\qquad(13)$$

with the integrated dynamical momentum of Eq. 1,

$$\Pi(t) = \hbar v_F \int_0^t \left(k_x + \frac{eE_0 t'}{\hbar} + \frac{eE_\mathrm{d}}{\hbar\omega_\mathrm{d}}\cos(\omega_\mathrm{d} t')\right)dt,\qquad(14)$$

such that we find the full transition probability

$$P(k_x, k_y) \approx \exp\left\{-\frac{\pi\hbar v_F k_y^2}{eE_0}\sum_{n\in\mathbb{Z}} J_n\left(\frac{-4v_F eE_\mathrm{d}}{\hbar\omega_\mathrm{d}^2}\sin\left(n\frac{\hbar\omega_\mathrm{d}^2}{4v_F eE_0}\right)\right)e^{in\hbar\omega_\mathrm{d}\frac{k_x}{eE_0}}\right\},\qquad(15)$$

for values of $k_x$ that are negative and large enough for the initial state to be in equilibrium prior to the quench transition. $J_n$ is the $n$th Bessel function of the first kind. This expression explicitly displays the periodicity of $eE_0\hbar^{-1}2\pi\omega_\mathrm{d}^{-1}$ in $k_x$ that is induced by the AC field. For the total current we average the transition probability over this periodicity and to leading order

find the expression

$$\bar{P}(k_y) = \int_0^{\frac{eE_0 2\pi}{\hbar \omega_d}} P(k_x, k_y) dk_x \approx c_0 + \frac{\pi^2 \hbar^2 v_F^2 k_y^4}{e^2 E_0^2} \sum_{n \in \mathbb{Z}} J_n^2 \left( \frac{-4v_F e E_d}{\hbar \omega_d^2} \sin \left( n \frac{\hbar \omega_d^2}{4 v_F e E_0} \right) \right), \quad (16)$$

where we ignore $c_0$ as it is constant with respect to $E_d$ and does not contribute to the photo-conductivity (See App. A). In Fig. 4 (c), we show the resulting contribution to the differential photoconductivity for small $k_y$

$$\tilde{g}_{xx}^{\text{LZ}} \propto \frac{d^2}{dE_d dE_0} \frac{\pi^2 \hbar v_F^2}{4 e \Gamma E_0} \sum_{n \in \mathbb{Z}} J_n^2 \left( \frac{-4 v_F e E_d}{\hbar \omega_d^2} \sin \left( n \frac{\hbar \omega_d^2}{4 v_F e E_0} \right) \right). \quad (17)$$

In the case of $E_0 \gg E_d$, $\tilde{g}_{xx}^{\text{LZ}}$ displays a striped structure that is consistent with the differential conductivity in Fig. 2 (a). For values of $2\pi E_0 \gtrsim E_d$, the prediction starts to deviate as the contributions from momenta with larger $k_y$ component are not captured by the leading order calculation for the transition probability in the modified LZ quench. The oscillating terms proportional to $E_0^{-1}$ in Eq. 17 result in erratic behavior for small DC fields, which leads to the chaotic results in the regime of $2\pi E_0 < E_d$. This demonstrates that the modified LZ quench is not a good description in the limit of $E_0 \ll E_d$, where many driving oscillations happen during the transition and dissipation cannot be neglected as a driven steady state forms.

In Figs. 4 (a) and (b) we show the time-evolution of the current density $j_x(t)$ as a function of the initial value of $k_x$ and for $v_F k_y = \omega_d/4$. We show this for $E_0 = 3\text{MV m}^{-1}$ and $E_d = 4.75\text{MV m}^{-1}$. Fig. 4 (a) shows the analytical solution of Eq. 13 to leading order in $k_y$, whereas Fig. 4 (b) shows the numerical simulation in the presence of weak dissipation. The dashed line indicates the points in time at which the drift of the momentum mode passes the band gap. Note the periodicity in the transition probability pattern as a function of $k_x$. The results agree very well with each other and display how dissipation acts as simple decay when the drift occurs quickly relative to the dissipation time scales. For larger values of $E_d$, momentum modes with large transversal components contribute to the current. The patterns at these large values of $k_y$ are not captured by the approximation to leading order in $k_y$. This explains the discrepancies of the striped patterns in Fig. 4 (c) and Fig. 2 (a) for increasing values of $E_d$.

In the limit of vanishing driving, i.e. $E_d \to 0$, the expression in Eq. 15 reproduces the well-known approximation of the transition probability of the LZ problem $P(k_y) \propto \exp\{-\pi \hbar v_F k_y^2 e^{-1} E_0^{-1}\}$. In this limit, we calculate the bare, i.e. undriven, non-linear conductivity of graphene for large $E_0$ (See App. A) and find

$$G_{xx}^{E_0 \gg 0} = \frac{dJ_x|_{E_d=0}}{dE_0} \approx 6 \sqrt{\frac{eE_0 v_F}{(\gamma_{\text{bg}} + \frac{1}{2}\gamma_-)^2 \hbar \pi^2}} \frac{e^2}{h}. \quad (18)$$

In the undriven, i.e. $E_d \to 0$, and linear limit, i.e. $E_0 \to 0$, we calculate the conductivity as a function of temperature (See App. A) and find

$$G_{xx}^{E_0 \to 0} \approx \frac{e^2}{h} \left( \frac{\pi}{2} - \arctan \left( \left( \frac{k_B T}{\hbar \gamma_1} \right)^{\frac{3}{4}} \right) + \frac{k_B T}{\hbar} \frac{\log(4)}{\gamma_-} \right). \quad (19)$$

A linear dependence of the minimal conductivity on the temperature in graphene as we find here is consistent with the literature [17–23, 28, 29]. In gated graphene this behavior reverses and the conductivity is found to decrease as a function of temperature [15, 24]. In the case of $T = 0$, Eq. 19 recovers the analytical result of the minimal conductivity of graphene

$$G_{xx}^{E_0 \to 0} \big|_{T \to 0} = \frac{\pi}{2} \frac{e^2}{h}. \quad (20)$$

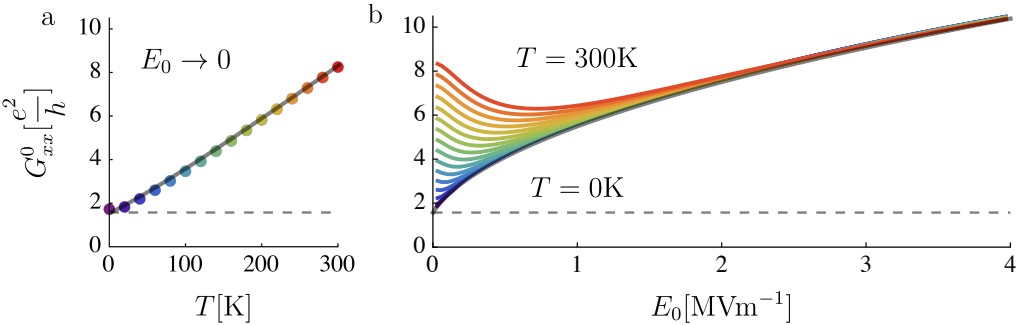

Figure 5: The undriven differential conductivity $G_{xx}^0$ of graphene for various temperatures. Panel (a) shows the linear conductivity as a function of temperature. The black line indicates the analytical solution in Eq. 19 (See App. A). Panel (b) shows the non-linear differential conductivity as a function of $E_0$ for various temperatures. In the limit of large $E_0$ the conductivity becomes temperature-independent. The black line shows the analytical expression for the conductivity presented in Eq. 21. The dashed lines indicate the bare conductivity of $\frac{\pi}{2}\frac{e^2}{h}$.

The minimal conductivity of graphene has been the subject of many studies and theory commonly produces the values $\frac{\pi}{2}\frac{e^2}{h}$ [7–9, 26] or $\frac{4}{\pi}\frac{e^2}{h}$ [9–11], while experiments consistently find $\frac{4e^2}{h}$.

In Fig. 5 we show the conductivity in the absence of any AC field, i.e. $G_{xx}^0$ as in Eq. 6, for different temperatures. In Fig. 5 (a) we show the linear conductivity, i.e. $E_0 \to 0$, as a function of temperature. Note that the expression in Eq. 19 is derived in the case of $\gamma_{\mathrm{bg}} = 0$. Therefore, we introduce a parameter $\gamma'$ as $\gamma_- \to \gamma_- + \gamma'$ and fit this to the numerical results of the Lindblad master equation. We find very good agreement for $\gamma' = \gamma_{\mathrm{bg}}/6$. The gray line in Fig. 5 (a) shows this fitted analytical prediction. In Fig. 5 (b) we show $G_{xx}^0$ as a function of $E_0$ for different temperatures. For small and intermediately large values of $E_0$ there is a clear dependency on the temperature. This is expected, since for large $E_0$ the current is dominated by contributions at momenta where the level spacing is large enough to suppress any thermal excitation. In general, the conductivity initially decreases with $E_0$, reaches a minimal value at some value of $E_0$ which increases with temperature. The conductivity then increases again while approaching the asymptotic behavior of Eq. 18. For $T = 300$K, the minimum of the differential conductivity is approximately located at $E_0 \approx 0.6$MV m$^{-1}$. This qualitative structure is consistent with recent results [71]. We combine the analytical results of Eqs. 18 and 19 into an expression for the differential conductivity at $T = 0$K

$$G_{xx}^0 \approx \frac{e^2}{h}\left( \alpha \frac{\pi}{2} + \sqrt{\frac{(1-\alpha)^2\pi^2}{4} + \frac{36eE_0 v_F}{(\gamma_{\mathrm{bg}} + \frac{1}{2}\gamma_-)^2\hbar\pi^2}} \right), \qquad (21)$$

where the construction of the parameter $\alpha$ ensures that for vanishing $E_0$ the numerical and analytical result is recovered, i.e. $G_{xx}^0|_{E_0 \to 0} = \frac{\pi}{2}\frac{e^2}{h}$ as in Eq. 20. We find very good agreement with the numerical results for $\alpha = \frac{1}{4}$ and show this estimate in Fig. 5 (b) as a black line. Note that this scaling behavior can in principle be used to determine the scale of dissipation $\Gamma$ in a given graphene sample. Note that the square-root scaling of Eq. 21 implies a power-law $\propto V^{\frac{3}{2}}$ in the $I$-$V$ curve of graphene, which is consistent with the non-linearities found in experimental measurements [72–74]. The expression in Eq. 21 for the differential conductivity is necessary in the context of the Tien-Gordon effect [68] as we discuss later.

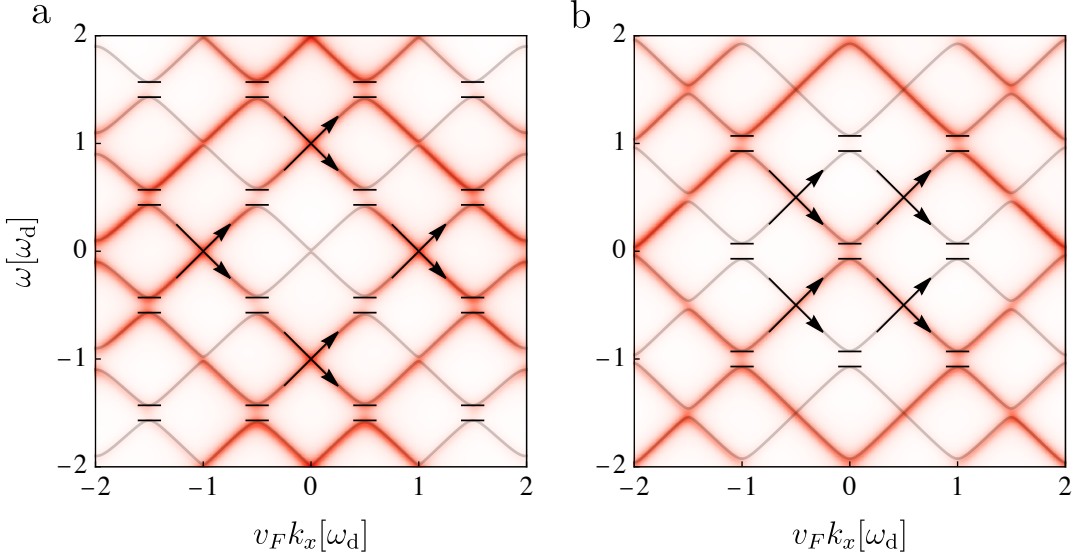

Figure 6: The electron distribution $n(\mathbf{k}, \omega)$ (See App. C) as a function of $k_x$ for $k_y = \omega_d/4v_F$ and for the AC field strengths $E_d = 10\,\mathrm{MV\,m^{-1}}$ (a) and $E_d = 13\,\mathrm{MV\,m^{-1}}$ (b). The solid lines show the Floquet spectra of the Hamiltonian in Eq. 24. Here, all dissipation coefficients are reduced by a factor of five compared to the values in the main text, for visual clarity. For increasing dissipation the gaps become increasingly indiscernible.

## 3.2 Dominant AC field

Here we analyze the regime in which the AC field is dominant, and the differential photoconductivity displays a checkerboard pattern as we show in Fig. 2 (a). As the AC field strength $E_d$ greatly exceeds the DC field strength $E_0$, we describe the system as primarily driven periodically and perturbed by the comparatively weak DC field $E_0$. This naturally suggests describing the system in the Floquet picture, in which dynamics are captured via the effective Floquet Hamiltonian

$$
H_F = \begin{pmatrix}
\ddots & & & & \\
& H_0 + \hbar\omega_d & H_1 & H_2 & \\
& H_{-1} & H_0 & H_1 & \\
& H_{-2} & H_{-1} & H_0 - \hbar\omega_d & \\
& & & & \ddots
\end{pmatrix}, \tag{22}
$$

which has an effective band structure that deviates from the undriven Hamiltonian. Here $H_m$ is the $m$th Fourier component of the original Hamiltonian $H(t)$ such that

$$
H_m = \frac{\omega_d}{2\pi} \int_0^{\frac{2\pi}{\omega_d}} e^{im\omega_d t} H(t) dt. \tag{23}
$$

The structure of the Floquet Hamiltonian explicitly includes infinitely many replicas of the bare bands coupled to each other by the components $H_{m\neq 0}$ and separated by multiples of the

photon energy $\hbar\omega_d$. In the case of Eq. 1 it is $H_{m,|m|>1} = 0$, and we explicitly write

$$
\frac{H_F}{\hbar} =
\begin{pmatrix}
\ddots & & & & & & \\
& \omega_d & v_F\bar{\mathbf{k}} & 0 & \Omega & 0 & 0 \\
& v_F\mathbf{k} & \omega_d & \Omega & 0 & 0 & 0 \\
& 0 & \Omega & 0 & v_F\bar{\mathbf{k}} & 0 & \Omega \\
& \Omega & 0 & v_F\mathbf{k} & 0 & \Omega & 0 \\
& 0 & 0 & 0 & \Omega & -\omega_d & v_F\bar{\mathbf{k}} \\
& 0 & 0 & \Omega & 0 & v_F\mathbf{k} & -\omega_d \\
& & & & & & \ddots
\end{pmatrix},
\tag{24}
$$

with $\mathbf{k} = k_x + ik_y$ and $\Omega = \frac{ev_F E_d}{2\hbar\omega_d}$.

In the case of very large AC field strengths, the Floquet band structure approaches an increasingly regular pattern in which the Floquet band gaps align with the quasi-resonant condition of $v_F k_x = m\omega_d/2$, $m \in \mathbb{Z}$. In Fig. 6 (a) and (b), we show the momentum- and frequency-resolved electron distributions $n(\mathbf{k}, \omega)$ (See App. C) at $v_F k_y = \omega_d/4$ for $E_d = 10\,\mathrm{MV\,m^{-1}}$ and $E_d = 13\,\mathrm{MV\,m^{-1}}$, respectively. We also show the eigenvalues of the Floquet Hamiltonian in Eq. 24 for the same parameters as solid lines. Note that for $E_d = 0$ the gap at $k_x = 0$ in this example is $\Delta = \hbar\omega_d/2$, which is fully suppressed in the effective Floquet band structure in Fig. 6. We calculate the Floquet energies at the quasi-resonant conditions in first order of $k_y$ (See App. B) and find that the $m$th Floquet band gap is given by

$$
\Delta\epsilon^{(m)} = 2\hbar v_F k_y J_m\left(2\frac{ev_F E_d}{\hbar\omega_d^2}\right) + \mathcal{O}(k_y^2),
\tag{25}
$$

where $J_m$ is the $m$th Bessel function of the first kind. For increasing $E_d$, the range of values of $k_y$ for which this expression remains valid increases. For those values of $E_d$ for which the $m$th Bessel function evaluates to zero, the transition probability becomes unity as a form of coherent destruction of tunneling [63, 66].

Fig. 2 (a) shows that the differential photoconductivity displays a type of checkerboard pattern in the regime that we consider in this section. The regularity of this pattern with respect to $E_0$ aligns along values of

$$
E_0^{(m)} = \frac{m\hbar\omega_d^2}{4\pi ev_F},
\tag{26}
$$

with $m \in \mathbb{N}$, as we indicate with vertical lines. These values of $E_0^{(m)}$ are the DC field strengths for which during one driving period $\frac{2\pi}{\omega_d}$, a shift in momentum equivalent to the difference in location between $m$ resonances is accumulated. In consideration of the Floquet band gaps in Eq. 25, this translates into a momentum shift that is commensurate with $m$ Floquet band gaps for small $k_y$.

We demonstrate that the contributions to the photoconductivity are located at these Floquet band gaps in Fig. 7 (a), where we show the photocurrent density $\Delta j_x(\mathbf{k})$, as defined in Eq. 10, integrated over $k_y$ as a function of $k_x$, and the AC field strength $E_d$ in the limit of small $E_0$. We see a clear structure of dominant contributions to the differential photoconductivity at momentum components $k_x = m\omega_d/v_F$, i.e. at locations of the Floquet band gaps for small $k_y$ as given in Eq. 25. With increasing $E_d$, contributions at higher-order resonances start to emerge that are aligned with the locations of Floquet band gaps. The regularity in the differential photoconductivity $g_{xx}$ with respect to $E_d$ emerges as a consequence of these quasi-resonant contributions, which relate to the size of the Floquet band gaps in Eq. 25. For

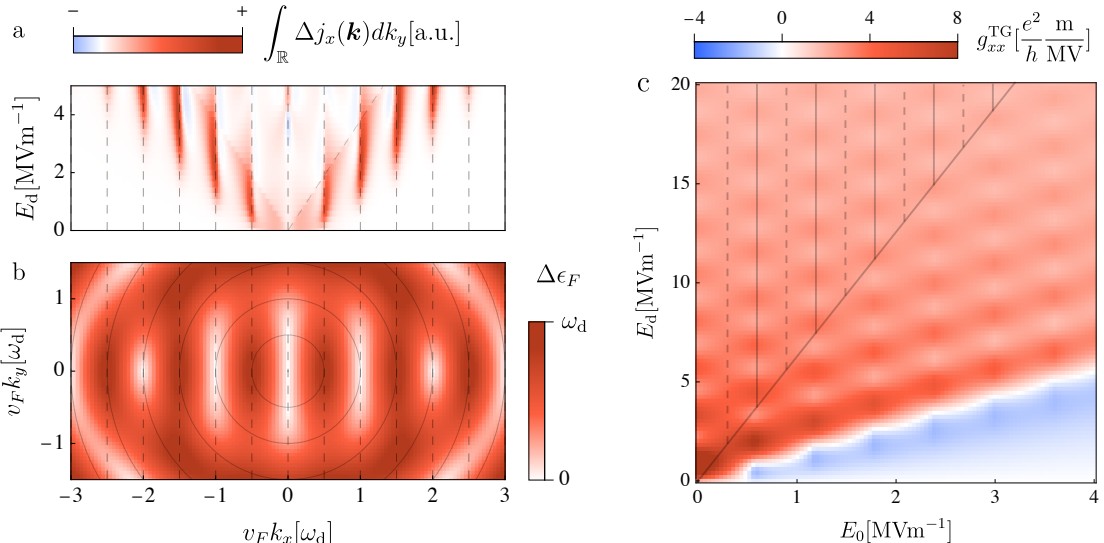

Figure 7: The distribution of current contributions in momentum space. In panel (a), we show the photocurrent density in momentum space integrated over only the $k_y$ component as a function of the $k_x$ component, and the AC field strength $E_d$. The dominant contributions are located at the multi-photon resonances indicated by the vertical dashed lines. In panel (b), we show the Floquet band energy differences resolved in $k$-space for large AC field strength $E_d = 10\text{MV m}^{-1}$. The Floquet gap locations align with the resonance conditions for $k_y = 0$, giving rise to a striped pattern of Floquet band gaps. The circles indicate the original resonance conditions in bare graphene. In panel (c), we show the differential photoconductivity as given by the Tien-Gordon expression in Eq. 28 using the expression for the bare differential conductivity in Eq. 21. Note the similarities in the regular checkerboard pattern to the differential photoconductivity in Fig. 2.

large values of $E_d$ the roots of the even-order Bessel functions in Eq. 25 become increasingly aligned, as do the roots of the odd-order Bessel functions. This results in an odd-even pattern of dominant contributions with respect to $E_0^{(m)}$, i.e. the checkerboard pattern in Fig. 2 (a). In Fig. 7 (b), we show the Floquet band energy differences for $E_d = 10\text{MV m}^{-1}$, where it is clearly visible how the Floquet band gaps for small $k_y$ align in the regular pattern along $k_x$ at which the dominant contributions to the photocurrent occur. This is consistent with pulsed two-level systems, in which LZ transitions across instantaneous Floquet band structures provide an accurate description of the dynamics [75].

For comparison, we consider the predictions of the Tien-Gordon effect. In semiconducting nanostructures that are AC and DC biased simultaneously, the DC $I$-$V$ curves relate to those of the same structures without an AC driving field taking into account photon-assisted tunneling. The original discussion put forth by Tien and Gordon discussed the dynamics of superconducting junctions in a perpendicular electric field [68]. For a bias voltage $V(t) = V_0 + \frac{V_d}{\omega_d}\sin(\omega_d t)$ they approximate the driven $I$-$V$ curve as

$$I(V_0) = \sum_m J_m^2\left(\frac{V_d}{\omega_d}\right)I_0(V_0 + m\omega_d). \tag{27}$$

A similar analysis has been put forth for graphene in the presence of an oscillating chemical potential [76], i.e. an alternating gate bias. The expression in Eq. 27 translates into the

conductivity

$$G_{xx}^{\mathrm{TG}} = \sum_m J_m^2 \left( \frac{e v_F E_{\mathrm{d}}}{\hbar \omega_{\mathrm{d}}^2} \right) G_{xx}^0 \left( E_0 + \frac{m \hbar \omega_{\mathrm{d}}^2}{2\pi v_F e} \right). \tag{28}$$

In Fig. 7 (c) we show the differential photoconductivity as it is predicted by Eq. 28 using the approximate expression for the bare conductivity in Eq. 21. We find that the characteristic structure shows a similar checkerboard pattern to that in Fig. 2 (a). In particular, the regularity with respect to $E_0$ is identical, as in Eq. 28 the same multi-photon offset to the DC field strength $E_0$ occurs as in Eq. 26. The regularity with respect to $E_{\mathrm{d}}$ agrees with the results in Fig. 2 (a) and the Floquet band gap scaling in Eq. 25, despite the distinct nature of the two phenomena. Note that the Tien-Gordon prediction in Eq. 28 becomes increasingly inaccurate towards the regime where it is $2\pi E_0 > E_{\mathrm{d}}$, which is where the checkerboard pattern is lost in the differential photoconductivity in Fig. 2 (a).

## 4 Conclusion

We have presented the non-linear longitudinal differential DC photoconductivity of light-driven monolayer graphene. We model the electron dynamics near the Dirac points of the graphene dispersion, in the presence of a DC bias field and an AC driving field. The driving field has the same linear polarization as the bias field. The differential photoconductivity displays two regimes as a function of the AC driving field strength, and the DC bias field strength, depending on which of these field strengths is dominant. The dynamics of these two regimes are captured well in two very distinct pictures, which we have explored analytically, and compared to the numerical result.

In the regime in which the DC bias field dominates, the photoconductivity displays a striped pattern. It derives from the Landau-Zener dynamics of the electrons across the Dirac cone, produced by the two electric fields. We presented an analytical calculation that reproduces the formation of the striped patterns in the photoconductivity. We put forth an analytical prediction for the formation of current density patterns in momentum space in the limit of large DC bias field strengths. From this we have approximated the non-linear conductivity in the absence of an alternating driving field, which shows behavior that is independent of temperature. Further, in the limit of a small DC bias and no driving, i.e. the undriven linear case, we have calculated that the minimal conductivity increases linearly with the temperature and recovers the value of $\frac{\pi}{2} \frac{e^2}{h}$ for vanishing temperature.

In the regime of the dominant AC driving field, the photoconductivity displays a checkerboard pattern. In this regime, we use a Floquet picture to describe the dynamics. The regularity as a function of the DC bias field is given by values which during one driving period accumulate a shift in momentum that is equal to the differences of Floquet band gap locations. We have shown that for large AC driving field and small transverse momenta, the Floquet band gaps align along the corresponding values of equal longitudinal momenta at which the dominant contributions to the photoconductivity are located. Further we have compared these results to the Tien-Gordon effect. We have used the analytic expression of the non-linear conductivity to calculate the differential photoconductivity using the expression put forth by Tien and Gordon, which displays a similar checkerboard pattern. In particular, the regularity as a function of the DC bias is manifestly identical. The regularity as a function of the driving field strength is very similar, and agrees with the analytic expression of the Floquet band gaps to leading order in transverse momentum.

The results we have put forth provide insight into the non-linear electronic transport properties of strongly driven graphene. The insights presented here support the engineering of non-equilibrium quantum electronic devices in the future.

## Acknowledgments

**Funding information** This work is funded by the Deutsche Forschungsgemeinschaft (DFG, German Research Foundation) – SFB-925 – project 170620586, and the Cluster of Excellence 'Advanced Imaging of Matter' (EXC 2056), Project No. 390715994.

## A Analytical approaches to the differential photoconductivity

### Transition probability for generalized quenches

We consider a two-level Hamiltonian with energy spacing $\Delta$ and a time-dependent quench $\pi(t)$ that we write as

$$H = \Delta\sigma_x + \pi(t)\sigma_z, \tag{A.1}$$

such that the second time-derivative of any state is given by

$$\partial_t^2 |\psi\rangle = -\frac{i}{\hbar}\partial_t(H|\psi\rangle) = -\frac{1}{\hbar}\left(\frac{1}{\hbar}\epsilon(t)^2 + i\sigma_z\dot{\pi}(t)\right)|\psi\rangle, \tag{A.2}$$

where we have used that $H^2 = \epsilon^2 = \Delta^2 + \pi(t)^2$ in centered two-level systems, i.e. when $\mathrm{Tr}(H) = 0$. For the components of $|\psi\rangle = (\psi_+, \psi_-)$ it is

$$\ddot{\psi}_\pm = -\frac{1}{\hbar^2}(\Delta^2 + \pi(t)^2 \pm i\hbar\dot{\pi}(t))\psi_\pm, \tag{A.3}$$

$$\dot{\psi}_\pm = -\frac{i}{\hbar}(\pm\pi(t)\psi_\pm + \Delta\psi_\mp), \tag{A.4}$$

such that for initial conditions $\psi_\pm(0)$ we have

$$\dot{\psi}_\pm(0) = -\frac{i}{\hbar}(\pm\pi(0)\psi_\pm(0) + \Delta\psi_\mp(0)). \tag{A.5}$$

We write $\psi_\pm(t) = \exp\{\xi_\pm(t)\}$ such that Eq. A.3 yields

$$\ddot{\xi}_\pm + \dot{\xi}_\pm^2 = -\frac{\Delta^2}{\hbar^2} - \frac{\pi^2}{\hbar^2} \mp \frac{i\dot{\pi}}{\hbar}, \tag{A.6}$$

$$\xi_\pm(0) = \ln(\psi_\pm(0)), \tag{A.7}$$

$$\dot{\xi}_\pm(0) = -\frac{i}{\hbar}\left(\pm\pi(0) + \Delta\frac{\psi_\mp(0)}{\psi_\pm(0)}\right). \tag{A.8}$$

We expand $\xi = \sum_{m=0}^{\infty} \xi_m \Delta^m$ in orders of $\Delta$ and write

$$\ddot{\xi}_{\pm,m} + \sum_{n=0}^{m} \dot{\xi}_{\pm,m-n}\dot{\xi}_{\pm,n} = -\delta_{m,2} - \delta_{m,0}(\pi^2 \pm i\dot{\pi}). \tag{A.9}$$

The case $m = 0$ gives

$$\ddot{\xi}_{\pm,0} + \dot{\xi}_{\pm,0}^2 = \mp i\dot{\pi} - \pi^2 \implies \dot{\xi}_{\pm,0} = \mp i\pi \tag{A.10}$$

$$\implies \xi_{\pm,0} = \ln(\psi_\pm(0)) \mp i\Pi(t), \tag{A.11}$$

with

$$\Pi(t) = \int_0^t \pi(t')dt'. \tag{A.12}$$

The case $m = 1$ is solved to satisfy the initial condition Eq. A.8 such that

$$\ddot{\xi}_{\pm,1} \mp i2\pi\dot{\xi}_{\pm,1} = 0 \implies \dot{\xi}_{\pm,1} = -i\frac{\psi_\mp}{\psi_\pm}e^{\pm 2\frac{i}{\hbar}\Pi(t)} \tag{A.13}$$

$$\implies \xi_{\pm,1} = -i\frac{\psi_\mp}{\psi_\pm}\int_0^t e^{\pm 2\frac{i}{\hbar}\Pi(t')}dt'. \tag{A.14}$$

The case $m = 2$ is solved as

$$\ddot{\xi}_{\pm,2} \mp i2\pi\dot{\xi}_{\pm,2} = -1 - \left(-i\frac{\psi_\mp}{\psi_\pm}e^{\pm 2\frac{i}{\hbar}\Pi(t)}\right)^2 \tag{A.15}$$

$$\implies \dot{\xi}_{\pm,2} = -e^{\pm i2\Pi(t)}\int_0^t e^{\mp i2\Pi(t')} - \frac{\psi_\mp^2}{\psi_\pm^2}e^{\pm 2i\Pi(t')}dt' \tag{A.16}$$

$$\implies \xi_{\pm,2} = \underbrace{-\int_0^t e^{\pm i2\Pi(t')}\int_0^{t'} e^{\mp i2\Pi(t'')}dt''dt'}_{\xi'_{\pm,2}} + \underbrace{\frac{\psi_\mp^2}{\psi_\pm^2}\int_0^t e^{\pm i2\Pi(t')}\int_0^{t'} e^{\pm i2\Pi(t'')}dt''dt'}_{\xi''_{\pm,2}}. \tag{A.17}$$

We note that $\xi'_{\pm,2} = \xi'^*_{\mp,2}$ such that we define $\xi'_2 = \text{Re}[\xi'_{+,2}] = \text{Re}[\xi'_{-,2}]$.
Higher orders of $m > 2$ are solved iteratively by

$$\ddot{\xi}_{\pm,m} \mp i2\pi\dot{\xi}_{\pm,m} = -\sum_{n=1}^{m-1}\dot{\xi}_{\pm,m-n}\dot{\xi}_{\pm,n} \tag{A.18}$$

$$\implies \dot{\xi}_{\pm,m} = -e^{\pm i2\Pi(t)}\int_0^t e^{\mp i2\Pi(t')}\sum_n\dot{\xi}_{\pm,m-n}\dot{\xi}_{\pm,n}dt' \tag{A.19}$$

$$\implies \xi_{\pm,m} = -\int_0^t e^{\pm i2\Pi(t')}\int_0^{t'} e^{\mp i2\Pi(t'')}\sum_n\dot{\xi}_{\pm,m-n}\dot{\xi}_{\pm,n}dt''dt'. \tag{A.20}$$

Here, we consider $\xi_-$ up to second order with the boundary conditions $\psi_-(0) = 1$ and $\psi_+(0) = 0$. We calculate the amplitude square

$$|a(t)|^2 = \exp\left\{\left(\xi_{-,0} + \xi_{-,2}\frac{\Delta^2}{\hbar^2}\right)\right\}\exp\left\{\left(\bar{\xi}_{-,0} + \bar{\xi}_{-,2}\frac{\Delta^2}{\hbar^2}\right)\right\} = \exp\left\{2\frac{\Delta^2}{\hbar^2}\text{Re}[\xi'_2]\right\} \tag{A.21}$$

$$= \exp\left\{-\frac{\Delta^2}{\hbar^2}\int_0^t\left(\cos\left(\frac{2}{\hbar}\Pi(t')\right)\int_0^{t'}\cos\left(\frac{2}{\hbar}\Pi(t'')\right)dt''\right.\right.$$
$$\left.\left. + \sin\left(\frac{2}{\hbar}\Pi(t')\right)\int_0^{t'}\sin\left(\frac{2}{\hbar}\Pi(t'')\right)dt''\right)dt'\right\} \tag{A.22}$$

$$= \exp\left\{-\frac{\Delta^2}{\hbar}\left(\left[\int_0^t\cos\left(\frac{2}{\hbar}\Pi(t')\right)dt'\right]^2 + \left[\int_0^t\sin\left(\frac{2}{\hbar}\Pi(t')\right)dt'\right]^2\right)\right\} \tag{A.23}$$

$$= \exp\left\{-\frac{\Delta^2}{\hbar}\left|\int_0^t e^{i\frac{2}{\hbar}\Pi(t')}dt'\right|^2\right\}. \tag{A.24}$$

We apply this to the explicit case of

$$\pi(t) = \hbar v_F k_x + e v_F E_0 t + \frac{e v_F E_d}{\omega_d}\sin(\omega_d t), \tag{A.25}$$

$$\Pi(t) = \hbar v_F k_x t + \frac{e v_F E_0 t^2}{2} - \frac{e v_F E_d}{\omega_d^2}\cos(\omega_d t) + \frac{e v_F E_d}{\omega_d^2}. \tag{A.26}$$

This constitutes a Landau-Zener quench with an additional alternating motion. The construction of the parameters is chosen to invoke the similarities to graphene. In particular, we write $\Delta = \hbar v_F k_y$ to emphasize this connection. Note that the Hamiltonian in Eq. A.1 then becomes equivalent to that in Eq. 1 under a simple basis rotation around $\sigma_y$. We continue by writing the central exponential function as

$$\exp\left\{i\frac{2}{\hbar}\Pi(t)\right\} = e^{i2\frac{ev_F E_d}{\hbar\omega_d^2}} e^{i2v_F k_x t} e^{\frac{i}{\hbar}ev_F E_0 t^2} e^{-i\frac{2ev_F E_d}{\hbar\omega_d^2}\cos(\omega_d t)} \tag{A.27}$$

$$= e^{i2\frac{ev_F E_d}{\hbar\omega_d^2}} \sum_{n\in\mathbb{Z}} i^n J_n\left(\frac{-2ev_F E_d}{\hbar\omega_d^2}\right) e^{i(n\omega_d+2v_F k_x)t} e^{\frac{i}{\hbar}ev_F E_0 t^2}. \tag{A.28}$$

We integrate this product of exponential functions for a fixed value of $n$, which gives

$$\int_0^t e^{i(n\omega_d+2v_F k_x)t'} e^{\frac{i}{\hbar}ev_F E_0 t'^2} dt' \tag{A.29}$$

$$= \sqrt{\frac{i\pi\hbar}{4ev_F E_0}} e^{-\frac{i\hbar(n\omega_d+2v_F k_x)^2}{4ev_F E_0}} \left(\mathrm{erf}\left(\sqrt{\frac{-i}{4E_0}}(n\omega_d+2k_x)\right) - \mathrm{erf}\left(\sqrt{\frac{-i}{4E_0}}(n\omega_d+2k_x+2E_0 t)\right)\right).$$

In order to consider the transition amplitude across the quench, we assume that $|k_x|$, with $k_x < 0$, and $t$ are both large enough such that the error functions approach $-1$ and $1$ for large times. Then the absolute square of the integral becomes

$$\left|\int_0^\infty \exp\{i2\Pi(t)\}dt\right|^2 \approx \frac{\hbar\pi}{ev_F E_0}\left|\sum_{n\in\mathbb{Z}} i^n J_n\left(\frac{-2ev_F E_d}{\hbar\omega_d^2}\right) e^{-\frac{i\hbar n^2\omega_d^2}{4ev_F E_0}} e^{-\frac{in\hbar\omega_d k_x}{eE_0}}\right|^2. \tag{A.30}$$

In the undriven case of $E_d = 0$ we correctly recover the first order correction to the transition amplitude as predicted in the Landau-Zener problem. In that case it is $J_0(-2E_d\omega_d^{-2}) = 1$ and $J_{n>0}(-2E_d\omega_d^{-2}) = 0$, such that

$$|a(t\to\infty)|^2 = e^{-\frac{\pi\hbar v_F k_y^2}{eE_0}}. \tag{A.31}$$

In the driven case the calculation is more intricate. Remember that the expression Eq. A.30 is periodic in $k_x$. We take the expression for the transition probability and simplify the absolute squared expression. It is

$$|a|^2 = \exp\left\{-\frac{\pi\hbar v_F k_y^2}{2eE_0}\left|\sum_{n\in\mathbb{Z}} i^n J_n\left(-\frac{2v_F eE_d}{\hbar\omega_d^2}\right) e^{-i\frac{\hbar n^2\omega_d^2}{4v_F eE_0}} e^{-i\frac{n\hbar\omega_d k_x}{eE_0}}\right|^2\right\} \tag{A.32}$$

$$= \exp\left\{-\frac{\pi\hbar v_F k_y^2}{2eE_0}\sum_{n,m\in\mathbb{Z}} i^{n-m} J_n\left(-\frac{2v_F eE_d}{\hbar\omega_d^2}\right) J_m\left(-\frac{2v_F eE_d}{\hbar\omega_d^2}\right) e^{-i\frac{\hbar(n^2-m^2)\omega_d^2}{4v_F eE_0}} e^{-i\frac{\hbar(n-m)\omega_d k_x}{eE_0}}\right\} \tag{A.33}$$

$$= \exp\left\{-\frac{\pi\hbar v_F k_y^2}{2eE_0}\sum_{n,m\in\mathbb{Z}} i^n J_{n+m}\left(-\frac{2v_F eE_d}{\hbar\omega_d^2}\right) J_m\left(-\frac{2v_F eE_d}{\hbar\omega_d^2}\right) e^{-i\frac{n\hbar(n+2m)\omega_d^2}{4v_F eE_0}} e^{-i\frac{n\hbar\omega_d k_x}{eE_0}}\right\} \tag{A.34}$$

$$= \prod_{n\in\mathbb{Z}} \exp\left\{-\frac{\pi\hbar v_F k_y^2}{2eE_0} J_n\left(-\frac{4v_F eE_d}{\hbar\omega_d^2}\sin\left(\frac{n\hbar\omega_d^2}{4v_F eE_0}\right)\right) e^{-i\frac{n\hbar\omega_d k_x}{eE_0}}\right\} \tag{A.35}$$

$$= e^{-\frac{\pi\hbar v_F k_y^2}{2eE_0}} \prod_{n=1}^\infty \exp\left\{-\frac{\pi\hbar v_F k_y^2}{eE_0} J_n\left(-\frac{4v_F eE_d}{\hbar\omega_d^2}\sin\left(\frac{n\hbar\omega_d^2}{4v_F eE_0}\right)\right) \cos\left(\frac{n\hbar\omega_d k_x}{eE_0}\right)\right\}, \tag{A.36}$$

where we have used that

$$i^n \sum_{m\in\mathbb{Z}} J_{n+m}\left(-\frac{2E_d}{\omega_d^2}\right) J_m\left(-\frac{2E_d}{\omega_d^2}\right) e^{-i\frac{n(n+2m)\omega_d^2}{4E_0}} = J_n\left(\frac{4E_d}{\omega_d^2}\sin\left(n\frac{\omega_d^2}{4E_0}\right)\right), \tag{A.37}$$

which we obtain by invoking the relations

$$J_\mu(z) = \frac{1}{2\pi} \int_0^{2\pi} e^{i\mu\tau - iz\cos(\tau)} d\tau \,, \tag{A.38}$$

$$J_\mu(z)J_\nu(z) = \frac{2}{\pi} \int_0^{\frac{\pi}{2}} J_{\mu+\nu}(2z\cos(t))\cos((\mu-\nu)t)dt \,. \tag{A.39}$$

In the following, we use the expression in Eq. A.36 to approximate contributions to the conductivity.

**Integrated contributions from modulated quench transitions**

We estimate the conductivity by first integrating the transition probability over momentum-space, which gives the total current. Afterwards we take the derivative with respect to the two bias strengths $E_0$ and $E_d$. However, that expression is only finite in the presence of dissipation which relaxes the excited states after the quench. Therefore, we consider this calculation in a two-step model, where the transition occurs in full and only afterwards dissipation starts to act, decaying any population that was excited. This is modeled by exponential decay by a phenomenological dampening coefficient $\Gamma$. Note that $k_x$ in $|a|^2$ is the initial value of the momentum component. We calculate the transition probability averaged over this initial value by writing

$$|a|^2 = \exp\left\{ -\frac{\pi\hbar v_F k_y^2}{2eE_0} \sum_{n\in\mathbb{Z}} J_n\left( -\frac{4v_F eE_d}{\hbar\omega_d^2} \sin\left( \frac{n\hbar\omega_d^2}{4v_F eE_0} \right) \right) e^{-i\frac{n\hbar\omega_d k_x}{eE_0}} \right\} \tag{A.40}$$

$$= \sum_{l=0}^{\infty} \frac{(-\frac{\pi\hbar v_F k_y^2}{2eE_0})^l}{l!} \sum_{\substack{\{n_j\in\mathbb{Z}\} \\ j\in\{1,\dots,l\}}} e^{-i\frac{(\sum_{j=1}^l n_j)\hbar\omega_d k_x}{eE_0}} \prod_{j=1}^{l} J_{n_j}\left( -\frac{4v_F eE_d}{\hbar\omega_d^2} \sin\left( \frac{n_j\hbar\omega_d^2}{4v_F eE_0} \right) \right) \,, \tag{A.41}$$

and integrating this expression to leading order such that

$$\overline{|a|^2} = \frac{\hbar\omega_d}{2\pi eE_0} \int_0^{\frac{2\pi eE_0}{\hbar\omega_d}} |a|^2 dk_{x,0} \tag{A.42}$$

$$= \sum_{l=0}^{\infty} \frac{\left(-\frac{\pi\hbar v_F k_y^2}{2eE_0}\right)^l}{l!} \sum_{\substack{\{n_j\in\mathbb{Z}\} \\ j\in\{1,\dots,l\}}} \delta(\sum_{j=1}^l n_j) \prod_{j=1}^{l} J_{n_j}\left( -\frac{4v_F eE_d}{\hbar\omega_d^2} \sin\left( \frac{n_j\hbar\omega_d^2}{4v_F eE_0} \right) \right) \tag{A.43}$$

$$= 1 - \frac{\pi\hbar v_F k_y^2}{2eE_0} + \left( \frac{\pi\hbar v_F k_y^2}{2eE_0} \right)^2 \sum_{n\in\mathbb{Z}} J_n^2\left( -\frac{4v_F eE_d}{\hbar\omega_d^2} \sin\left( \frac{n\hbar\omega_d^2}{4v_F eE_0} \right) \right) + \mathcal{O}(k_y^6) \,. \tag{A.44}$$

In the first step we evaluated the integral and found that the exponential functions lead to vanishing integration unless the exponent is zero. This condition is expressed through the delta-function $\delta(\sum_{j=1}^l n_j)$. In the following step we have evaluated the first three terms of the series. Note that the $l = 1$ term is proportional to $J_0(0) = 1$ and for the $l = 2$ term the sum reduces since $n_2 = -n_1$ and a sign change in both the index of the Bessel function and the sine in the argument of the same Bessel function cancel each other. As we are interested in the differential photoconductivity, we first notice that the first two terms vanish when differentiating with respect to $E_d$, such that we can ignore them. We consider the conductivity in the limit,

where the quench occurs fast and dissipation acts afterwards with a simple exponential decay such that we integrate current contributions over all $k_x$ as

$$\tilde{g}_{xx} \propto \partial_{E_d} \partial_{E_0} \overline{|a|^2} \int_0^\infty e^{-\Gamma \frac{\hbar k_x}{eE_0}} dk_x \tag{A.45}$$

$$= \partial_{E_d} \partial_{E_0} \left( \overline{|a|^2} \frac{eE_0}{\hbar\Gamma} \right) \tag{A.46}$$

$$\approx k_y^4 \frac{\pi^2 \hbar v_F^2}{4e\Gamma} \partial_{E_d} \partial_{E_0} \sum_{n\in\mathbb{Z}} \frac{1}{E_0} J_n^2 \left( -\frac{4v_F eE_d}{\hbar \omega_d^2} \sin\left( \frac{n\hbar\omega_d^2}{4v_F eE_0} \right) \right). \tag{A.47}$$

This expression provides the contributions to the differential photoconductivity as a function of $k_y$ for small values of $k_y$. In order to access higher order contributions it is necessary to evaluate Eq. A.20 and then perform the analogous calculation of the conductivity.

## Non-linear conductivity in the large-DC-bias limit

For large values of $e v_F E_0 \hbar^{-1} \gg \Gamma \omega_d$, we can approximate the transition probability of a ground state at $t \to -\infty$ towards $t \to \infty$ as a function of $k_y$. We assume that the transition happens fast enough such that the entire transition process happens and then decay starts acting. In a comoving frame, time translates into $k_x$-momentum and the transition probability is proportional to the conductivity density. We take the result for the transition probability for $E_d = 0$ and multiply this by an exponential decay in $k_x$ starting at $k_x = 0$. It is

$$|a(k_x, k_y)|^2 = e^{-\frac{\pi v_F \hbar k_y^2}{eE_0}} e^{-\Gamma \frac{\hbar k_x}{eE_0}} \Theta(k_x), \tag{A.48}$$

where $\Theta(k_x)$ is the Heaviside function. This integrates over momentum space to

$$J_x = n_s n_v \frac{e v_F}{4\pi^2} \int_{-\infty}^\infty \int_0^\infty 2|a(k_x, k_y)|^2 dk_x dk_y = 2\sqrt{\frac{e^5 E_0^3 v_F}{\Gamma^2 \hbar^3 \pi^4}}. \tag{A.49}$$

Hence the conductivity from this main contributions is

$$G_{xx}^0 = \frac{dJ_x}{dE_0} = 6\sqrt{\frac{eE_0 v_F}{\Gamma^2 \hbar \pi^2}} \frac{e^2}{h}. \tag{A.50}$$

This is in agreement with the large $E_0$ limit of Fig. 5 (b) up to some constant offset provided by the current density close to the Dirac point which we neglected here. This calculation holds only in the limit where the exponential decay in $k_x$-direction happens close enough to the Dirac point that a linear dispersion is still a valid approximation. From the Lindbladian in the $4 \times 4$ basis it is easy to see, that $\Gamma = \gamma_\pm + \gamma_{bg}$.

## Temperature-dependent conductivity without driving

In the limit of small $E_0$, dissipation acts significantly during the transition and the previous approximation is no longer valid. Also, temperature starts to become relevant in this limit. In the comoving frame $k_x \to k_x - eE_0 \hbar^{-1} t$, the solution is a stationary state such that $\hbar \partial_t \rho = eE_0 \partial_{k_x} \rho$. To approach this problem analytically we neglect the expanded sector of the Hilbert space and only consider the original $2 \times 2$ system, i.e. $\gamma_{bg} = 0$. In this case the Lindblad-von Neumann master equation can be cast into the form

$$\hbar \partial_t \vec{\rho} = 2\vec{H} \times \vec{\rho} - \hbar\gamma_1 \vec{\rho} - \hbar\gamma_2 \vec{h} - \hbar\gamma_3 (\vec{\rho}\vec{h})\vec{h}, \tag{A.51}$$

for the density operator representation

$$\rho = \frac{1}{2}(\mathbb{1} + \vec{\rho}\vec{\sigma}), \qquad (A.52)$$

where $\vec{\sigma} = (\sigma_x, \sigma_y, \sigma_z)$ is a vector containing the Pauli matrices. Similarly, it is

$$H = \vec{H}\vec{\sigma}, \qquad (A.53)$$

and $\vec{h} = \epsilon^{-1}\vec{H}$ with $\epsilon = \sqrt{\vec{H}.\vec{H}}$. For details on and a derivation of this representation of the master equation we refer to previous work [77]. We invoke this representation in the comoving frame and write

$$eE_0\partial_{k_x}\vec{\rho} = 2\vec{H} \times \vec{\rho} - \hbar\gamma_1\vec{\rho} - \hbar\gamma_2\vec{h} - \hbar\gamma_3(\vec{\rho}\vec{h})\vec{h}. \qquad (A.54)$$

In the case of finite temperature the dissipation coefficient $\gamma_2$ depends on the temperature $T$ as

$$\gamma_2 = \gamma_\pm \tanh\left(\frac{\epsilon}{k_B T}\right), \qquad (A.55)$$

where $\epsilon = \sqrt{(\hbar v_F k_x + e v_F E_0 t)^2 + (\hbar v_F k_y)^2}$. We consider this in the expansion in orders of $E_0$ and find

$$e\partial_{k_x}\vec{\rho}_{m-1} = 2(\vec{H}_0 \times \vec{\rho}_m + \vec{H}_1 \times \vec{\rho}_{m-1}) - \hbar\gamma_1\vec{\rho}_m - \hbar\gamma_{2,m}\vec{h}_{m-n} - \hbar\sum_{n,l}\gamma_3(\vec{\rho}_{m-n-l}\vec{h}_n)\vec{h}_l. \qquad (A.56)$$

We solve this for $m = 0$ and find the thermal state

$$\vec{\rho}_0 = -\tanh\left(\frac{\hbar v_F k}{k_B T}\right)\vec{h}. \qquad (A.57)$$

We continue to find the solution for $m = 1$, which expressed in polar coordinates $k_x + ik_y = ke^{i\phi}$, has the $x$-component

$$\rho_{1,x} = \frac{e}{2}\text{sech}^2\left(\frac{\hbar v_F k}{k_B T}\right)\left(\frac{2\hbar^2 v_F^2 \cos^2(\phi)}{\hbar^2 k_B T(\gamma_1 + \gamma_3)} + \frac{\gamma_1 \hbar^2 v_F^2 \sin^2(\phi)\sinh\left(\frac{2\hbar v_F k}{k_B T}\right)}{\hbar^2 \gamma_1^2 \hbar v_F k + 4\hbar^3 v_F^3 k^3}\right). \qquad (A.58)$$

In order to obtain the conductivity we have to integrate this expression over momentum space. This expression integrates angularly to

$$\frac{1}{e}\int_0^{2\pi}\rho_{1,x}d\phi = \frac{\pi\hbar^2 v_F^2\text{sech}^2\left(\frac{\hbar v_F k}{k_B T}\right)}{\hbar^2 k_B T(\gamma_1 + \gamma_3)} + \frac{\pi\hbar^2 v_F^2\gamma_1\tanh\left(\frac{\hbar v_F k}{k_B T}\right)}{\hbar^3 v_F k\gamma_1^2 + 4\hbar^3 v_F^3 k^3}. \qquad (A.59)$$

The first term integrates radially to

$$\int_0^\infty k\frac{\pi\hbar^2 v_F^2\text{sech}^2\left(\frac{\hbar v_f k}{k_B T}\right)}{\hbar^2 k_B T(\gamma_1 + \gamma_3)}dk = \frac{k_B T}{\hbar^2}\frac{\pi\log(2)}{\gamma_1 + \gamma_3}. \qquad (A.60)$$

The second term relies on some approximation of the hyperbolic tangent, we use to good approximation that

$$\int_0^\infty \frac{\pi\hbar^2 v_F^2\gamma_1\tanh\left(\frac{\hbar v_F k}{k_B T}\right)}{\hbar^3 v_F\gamma_1^2 + 4\hbar^3 v_F^3 k^2}dk \approx \left(\frac{\pi^2}{4} - \frac{\pi}{2}\arctan\left(\left(\frac{k_B T}{\hbar\gamma_1}\right)^{\frac{3}{4}}\right)\right)\frac{1}{\hbar v_F}. \qquad (A.61)$$

We collect the terms and write the conductivity as

$$G_{xx}^0 = n_v n_s \frac{e v_F}{4\pi^2} \int_0^\infty \int_0^{2\pi} k \rho_{1,x}(k) d\phi \, dk, \qquad (A.62)$$

with the valley- and spin-degeneracy $n_v = n_s = 2$. With this we insert the derived expressions and get the final result

$$G_{xx}^0 \approx \left( \frac{\pi}{2} \left( 1 - \frac{2}{\pi} \arctan\left( \left( \frac{k_B T}{\hbar \gamma_1} \right)^{\frac{3}{4}} \right) \right) + \frac{k_B T}{\hbar} \frac{2 \log(2)}{\gamma_1 + \gamma_3} \right) \frac{e^2}{h}. \qquad (A.63)$$

# B  Floquet band gaps at small transverse momenta

We want to calculate the Floquet band structure in the limit of small $k_y$ for the driven Hamiltonian

$$\frac{1}{\hbar} H = v_F \left( k_x + \frac{e E_d}{\hbar \omega_d} \cos(\omega_d t) \right) \sigma_x + v_F k_y \sigma_y. \qquad (B.1)$$

We start by going into the time-dependent basis under the transformation

$$V = \exp\left\{ -i v_F \left( k_x t + \frac{e E_d}{\hbar \omega_d^2} \sin(\omega_d t) \sigma_x \right) \right\}, \qquad (B.2)$$

such that the Schrödinger equation becomes

$$\partial_t \psi = -i v_F k_y V^\dagger \sigma_y V \psi, \qquad (B.3)$$

with the effective Hamiltonian proportional to

$$V^\dagger \sigma_y V = \sin\left( 2\left( v_F k_x t + \frac{e v_F E_d}{\hbar \omega_d^2} \sin(\omega_d t) \right) \right) \sigma_z + \cos\left( 2\left( v_F k_x t + \frac{e v_F E_d}{\hbar \omega_d^2} \sin(\omega_d t) \right) \right) \sigma_y. \qquad (B.4)$$

In this basis the time-evolution operator after one driving period describes the same transformation as the effective static Floquet Hamiltonian. The formal solution yields

$$U\left( \frac{2\pi}{\omega_d} \right) = \hat{T}[\exp\{ -i v_F k_y \int_0^{\frac{2\pi}{\omega_d}} V^\dagger \sigma_y V dt \}], \qquad (B.5)$$

where $\hat{T}$ indicates time-ordering. This expression is a formal notation for an infinite series of nested integrals. We approximate this transformation by the Magnus expansion to first order in $k_y$. It then suffices to calculate

$$U\left( \frac{2\pi}{\omega_d} \right) = \exp\{ -i v_F k_y \int_0^{\frac{2\pi}{\omega_d}} V^\dagger \sigma_y V dt \}, \qquad (B.6)$$

where the integral in the exponential is now a closed expression that can be integrated by itself

before exponentiation. We calculate the integral by first writing

$$\sin\left(2\left(v_F k_x t + \frac{e v_F E_d}{\hbar \omega_d^2}\sin(\omega_d t)\right)\right) = \frac{1}{2i}\sum_{n\in\mathbb{Z}} i^n e^{i2v_F k_x t} J_n\left(2\frac{e v_F E_d}{\hbar \omega_d^2}\right) e^{in(\omega_d t - \frac{\pi}{2})}$$

$$- i^n e^{-i2v_F k_x t} J_n\left(2\frac{e v_F E_d}{\hbar \omega_d^2}\right) e^{in(-\omega_d t - \frac{\pi}{2})} \tag{B.7}$$

$$= \sum_{n\in\mathbb{Z}} J_n\left(2\frac{e v_F E_d}{\hbar \omega_d^2}\right)\sin(2v_F k_x t - n\omega_d t) \tag{B.8}$$

$$\int_0^{\frac{2\pi}{\omega_d}} \to \frac{2\pi}{\omega_d}\sum_{n\in\mathbb{Z}} J_n\left(2\frac{e v_F E_d}{\hbar \omega_d^2}\right)\frac{1 - \cos\left(2\pi\left(\frac{2v_F k_x}{\omega_d} - n\right)\right)}{\frac{2v_F k_x}{\omega_d} - n}, \tag{B.9}$$

and analogously

$$\cos\left(2\left(v_F k_x t + \frac{e v_F E_d}{\hbar \omega_d^2}\sin(\omega_d t)\right)\right) \to \frac{2\pi}{\omega_d}\sum_{n\in\mathbb{Z}} J_n\left(2\frac{e v_F E_d}{\hbar \omega_d^2}\right)\frac{\sin\left(2\pi\left(\frac{2v_F k_x}{\omega_d} - n\right)\right)}{\frac{2v_F k_x}{\omega_d} - n}. \tag{B.10}$$

The final transformation in the original basis is

$$V^\dagger\left(\frac{2\pi}{\omega_d}\right) U\left(\frac{2\pi}{\omega_d}\right), \tag{B.11}$$

which has the eigenvalues $\lambda_\pm = e^{\pm i\epsilon \frac{2\pi}{\hbar \omega_d}}$, where $\epsilon$ are the Floquet energies. Hence, the Floquet energies can be calculated as

$$\epsilon_\pm = \frac{\hbar \omega_d}{i2\pi}\log(\lambda_\pm). \tag{B.12}$$

In the case in which $2v_F k_x \omega_d^{-1}$ is integer valued, we simplify

$$\frac{2\pi}{\omega_d}\sum_{n\in\mathbb{Z}} J_n\left(2\frac{e v_F E_d}{\hbar \omega_d^2}\right)\frac{1 - \cos(2\pi(l-n))}{l-n} = 0, \tag{B.13}$$

$$\frac{2\pi}{\omega_d}\sum_{n\in\mathbb{Z}} J_n\left(2\frac{e v_F E_d}{\hbar \omega_d^2}\right)\frac{\sin(2\pi(l-n))}{l-n} = \frac{2\pi}{\omega_d}J_{2l}\left(2\frac{e v_F E_d}{\hbar \omega_d^2}\right). \tag{B.14}$$

and similarly in the case where $2v_F k_x \omega_d^{-1}$ is half-integer we write

$$\frac{2\pi}{\omega_d}\sum_{n\in\mathbb{Z}} J_n\left(2\frac{e v_F E_d}{\hbar \omega_d^2}\right)\frac{1 - \cos(2\pi(l-n))}{l-n} = \frac{2\pi}{\omega_d}J_{2l+1}\left(2\frac{e v_F E_d}{\hbar \omega_d^2}\right), \tag{B.15}$$

$$\frac{2\pi}{\omega_d}\sum_{n\in\mathbb{Z}} J_n\left(2\frac{e v_F E_d}{\hbar \omega_d^2}\right)\frac{\sin(2\pi(l-n))}{l-n} = 0. \tag{B.16}$$

In either of these case it is $V(\frac{2\pi}{\omega_d}) \propto \mathbb{1}$ and we therefore find

$$U\left(\frac{2\pi}{\omega_d}\right) = e^{-i\frac{2\pi \hbar v_F k_y}{\hbar \omega_d} J_l\left(2\frac{e v_F E_d}{\hbar \omega_d^2}\right)\sigma_y}, \tag{B.17}$$

which has the eigenvalues

$$\lambda_\pm = e^{\mp i\frac{2\pi \hbar v_F k_y}{\hbar \omega_d} J_l\left(2\frac{e v_F E_d}{\hbar \omega_d^2}\right)}, \tag{B.18}$$

and we therefore find the Floquet energies

$$\epsilon_\pm = \hbar v_F k_y J_l\left(2\frac{e v_F E_{\mathrm{d}}}{\hbar \omega_{\mathrm{d}}^2}\right).\tag{B.19}$$

Note that $\frac{2v_F k_y}{\omega_{\mathrm{d}}} < 1$ and $J_l(z) \leq 1$, such that analytic continuation in the complex plane does not have to be considered and the eigenvalues take this simple shape.

## C Dissipative calculations

The model of graphene that we use in this work employs of a Hilbert space that includes two additional states. We consider fermionic operators $c_{\mathbf{k},A}^{(\dagger)}$ and $c_{\mathbf{k},B}^{(\dagger)}$ that annihilate (create) electrons at momentum $\mathbf{k}$ in sublattice $A$ and $B$, respectively. The Hilbert space is then spanned by the states $|0\rangle$, $c_A^\dagger|0\rangle$, $c_B^\dagger|0\rangle$ and $c_B^\dagger c_A^\dagger|0\rangle$. In particular, the first and fourth states describe the situation in which either no electrons or two electrons are occupying a given momentum mode $\mathbf{k}$. In this space we define the dissipation operators in the instantaneous eigenbasis of the driven Hamiltonian, e.g. $L_-$ describes decay from the excited single-electron eigenstate to the single-electron ground state. It is

$$L_- = \begin{pmatrix} 0 & 0 & 0 & 0 \\ 0 & 0 & 0 & 0 \\ 0 & 1 & 0 & 0 \\ 0 & 0 & 0 & 0 \end{pmatrix}, \qquad L_+ = \begin{pmatrix} 0 & 0 & 0 & 0 \\ 0 & 0 & 1 & 0 \\ 0 & 0 & 0 & 0 \\ 0 & 0 & 0 & 0 \end{pmatrix}, \quad L_z = \begin{pmatrix} 0 & 0 & 0 & 0 \\ 0 & 1 & 0 & 0 \\ 0 & 0 & -1 & 0 \\ 0 & 0 & 0 & 0 \end{pmatrix}, \quad \text{(C.1)}$$

$$L_{\downarrow,j} = \begin{pmatrix} 0 & \delta_{j,1} & 0 & 0 \\ 0 & 0 & 0 & 0 \\ \delta_{j,2} & 0 & 0 & \delta_{j,3} \\ 0 & \delta_{j,4} & 0 & 0 \end{pmatrix}, \quad L_{\uparrow,j} = \begin{pmatrix} 0 & 0 & \delta_{j,2} & 0 \\ \delta_{j,1} & 0 & 0 & \delta_{j,4} \\ 0 & 0 & 0 & 0 \\ 0 & 0 & \delta_{j,3} & 0 \end{pmatrix}.\tag{C.2}$$

$L_{\uparrow/\downarrow,j}$ describe the individual transitions in and out of the single-electron sector. In the master equation these particular processes are weighted by the dissipation rate $\gamma_{\mathrm{bg}} = \gamma_\downarrow + \gamma_\uparrow$ and $\gamma_\pm = \gamma_+ + \gamma_-$ We weight opposing processes by Boltzmann factors that ensure a Fermi distribution with temperature $T$ in the equilibrium state

$$\gamma_\uparrow = \gamma_\downarrow \exp\left\{-\frac{\epsilon}{k_B T}\right\},\tag{C.3}$$

$$\gamma_+ = \gamma_- \exp\left\{-\frac{2\epsilon}{k_B T}\right\}.\tag{C.4}$$

$k_B$ is the Boltzmann constant and $\pm\epsilon$ are the instantaneous eigenenergies of the Hamiltonian, e.g. Eq. 1. For further details on this dissipative model we refer to previous work that employ the same model [49, 69, 70].

Further, this description allows us to calculate two-point correlation functions using these fermionic operators, such as $\langle c_A^\dagger(t_2) c_A(t_1)\rangle$ and $\langle c_B^\dagger(t_2) c_B(t_1)\rangle$. We use these correlation functions to calculate the momentum- and frequency-resolved electron distribution in the steady state of the driven dissipative system.

$$n(\mathbf{k},\omega) = \frac{1}{(t_f - t_i)^2} \int_{t_i}^{t_f} \int_{t_i}^{t_f} \sum_{j=A,B} \langle c_{\mathbf{k},j}^\dagger(t_2) c_{\mathbf{k},j}(t_1)\rangle e^{i\omega(t_2 - t_1)} dt_1 dt_2.\tag{C.5}$$

This electron distribution reveals the Floquet band structure of the system and the electron population in these bands. This quantity is reminiscent of time- and angle-resolved photoelectron spectroscopy [78]. For further details on this method see previous works [49, 69, 70]. We use this calculation for Fig. 7 in the main text.

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
