# Peer review of "Non-linear photoconductivity of strongly driven graphene"

_SciPost Physics, doi:SciPost Phys. 18, 199 (2025)_

## Round 1 · Referee Report · Anonymous (Referee 1) · 2024-2-15

Report

The authors report the results of a theoretical study of graphene conductivity under the influence of strong ac and/or dc electric fields. The topic of this manuscript lies in the broad field of research on the nonlinear electromagnetic response of graphene, which began in 2007 after it was predicted that graphene should have highly nonlinear electrodynamic properties. The paper is potentially interesting and important but it is unclear whether the results of this work are new. There is an extensive literature on the relevant theoretical studies, first of all, the papers by Mikhailov and the group of Sipe, but the authors do not cite any of them and do not compare their results with those already available in the literature. Among the theoretical results obtained both within the framework of perturbation theory and non-perturbatively, both for the general third-order response and for the photoconductivity, the following works can be mentioned:
Mikhailov, Non-linear electromagnetic response of graphene, Europhys. Lett. 79, 27002 (2007)
Cheng et al., DC current induced second order optical nonlinearity in graphene, Optics Express 22 (13), 15868-15876 (2014)
Cheng et al., Third order optical nonlinearity of graphene, New Journal of Physics 16 (5), 053014 (2014)
Cheng et al., Third-order nonlinearity of graphene: Effects of phenomenological relaxation and finite temperature, Phys. Rev. B 91 (23), 235320 (2015)
Mikhailov, Quantum theory of the third-order nonlinear electrodynamic effects of graphene, Phys. Rev. B 93, 085403 (2016)
Mikhailov et al., Negative dynamic conductivity of a current-driven array of graphene nanoribbons, Phys. Rev. B 94, 035439 (2016)
Cheng et al., Numerical study of the optical nonlinearity of doped and gapped graphene: From weak to strong field excitation, Phys. Rev. B 92 (23), 235307
Al-Naib et al., Nonperturbative model of harmonic generation in undoped graphene in the terahertz regime, New Journal of Physics 17 (11), 113018 (2015)
Mikhailov, Nonperturbative quasiclassical theory of the nonlinear electrodynamic response of graphene, Phys. Rev. B 95, 085432 (2017)
Mikhailov, Theory of the strongly nonlinear electrodynamic response of graphene: A hot electron model, Phys. Rev. B 100, 115416 (2019)
Mikhailov, Nonperturbative quasiclassical theory of graphene photoconductivity”, Phys. Rev. B 103, 245406 (2021)
Authors should check whether their results are new compared to those already available in the literature, compare their and published results in the manuscript, and cite relevant articles accordingly. If there is anything new in the submitted manuscript, it may be published after the proposed changes have been made.
  • validity: -
  • significance: -
  • originality: -
  • clarity: -
  • formatting: -
  • grammar: -

Author:  Lukas Broers  on 2024-09-30  [id 4808]

(in reply to Report 1 on 2024-02-15)

We thank the referee for their positive assessment of our work. We appreciate the referee drawing our attention to this body of work that we had not referenced. After studying the provided literature, we agree that the works are largely related and relevant. These studies cover different aspects of the large research field of electromagnetic responses in graphene. They all differ from our work in that they describe optical rather than direct responses, that the driving frequency is small compared to the chemical potential, that they utilize a semi-classical Boltzmann approach, or that they focus on third-order susceptibilities, to name the main differences. In most cases all or most of these points apply. We therefore did not find any instances of overlap in the results that would take away from the novelty of our results, nor stand in contradiction to them.

Regardless, we agree that these works should be mentioned and put into context in the introduction of our manuscript. We have therefore cited the provided references in the revised manuscript. We note that some of these references were also brought up by the other referee.

---

## Round 1 · Referee Report · Anonymous (Referee 2) · 2024-2-19

Report

This is a single-blind review of SciPost Physics manuscript 2312.13217v1 titled "Non-linear photoconductivity of strongly driven graphene" by Lukas Broers and Ludwig Mathey. The manuscript is a 27-pages long PDF file, including 7 figures, 117 equations, a 62-entry bibliography and 3 appendices with mathematical calculations (equations 29-117). The same manuscript is also on arxiv, https://arxiv.org/abs/2312.13217.

The work presented in the manuscript consists of the theoretical analysis of the longitudinal photoconductivity produced when a graphene monolayer is subjected to an E-field with a strong DC and/or strong AC (harmonic THz) component; the E-field is parallel to graphene. The quantum system studied is open, i.e., it assumes various dissipation and dephasing mechanisms, introduced by corresponding rates. The Authors identify two distinct regimes (AC>>DC or AC<<DC), for which they extract analytical formulas identifying the relevant boundaries and evaluating the response, i.e., the differential photo/conductivity (change in electric current due to a strong AC/DC E-field presence), which is highly nonlinear. The analytical results match the numerical ones, nothing that some of the formulas contain heuristically introduced parameters which can be tuned to fit the numerically computed response.

The paper is in overall very well written and presented, scientifically correct (theory matches numerical computation) and is positioned in the area of 2D material high-frequency nonlinearities which is highly relevant to the analysis and design of non-equilibrium quantum-electronic/photonic devices. For these reasons, I recommend publication of this paper after minor improvements. Below follows a list of comments that the Authors can use to improve their work by extending its impact on areas beyond mathematical physics:

A/ Even though the introduction is rather clear about the contents of the paper, it should further highlight the novelty of its core findings/results, with respect to the relevant body of the work in the literature. For instance, one wonders if these findings are consistent with this work https://doi.org/10.1103/PhysRevB.103.245406

B/ The experimental audience of the journal would appreciate a linking of these theoretical results to measured ones, at the FIR frequency chosen (~10 THz) or elsewhere (e.g., at NIR/optical or low-THz bands). As the paper mentions, such a comparison can be potentially exploited in the characterization of graphene quality, e.g., in the extraction of the effective dissipation-rate parameters from measured spectra.

C/ The Authors could consider to compare their results to quantities relevant to nonlinear optics, such third-order/Kerr-like effect [:https://doi.org/10.1103/PhysRevB.91.235320] or saturable absorption regimes [https://doi.org/10.1103/PhysRevB.100.115416]. Such cases could arise by their model in the limiting case of zero DC component (only AC drive) and by including energy-dependent relaxation (dissipation) rates. Coherent nonlinearities, in the phase and/or the magnitude of AC photoconductivity, are highly relevant to graphene-comprising electro-optic applications such as lasers or photodiodes.

D/ In several places in the paper, the dissipation is termed "small" (and has been further reduced, in some figures, for visual clarity); this refers to all the dissipation rates employed or to some of them? Do the Authors expect a different picture (for the two regimes identified) for more highly dissipative systems? The latter could be a parametric study using the derived formulas. As with the previous comment, the study of complex valued (i.e., in both magnitude and phase) photoconductivity in such high-loss nonlinear systems is relevant to applications in coherent and non-Hermitian photonic systems and devices/components.

E/ Some minor issues: (i) The paper should be swiped for consistency in the terminology used, e.g., use "bias" only for the DC component and "drive" only the AC one; presently, the word "probe" is sometimes used for DC and the word "drive" is sometimes used for either AC or DC; there is a typo (AC-->DC) in the first line of section 3.1. (ii) The reference list should be checked as some titles contain inappropriate lowercase characters (e.g., in names like zener or in acronyms like thz).
  • validity: -
  • significance: -
  • originality: -
  • clarity: high
  • formatting: excellent
  • grammar: excellent

Author:  Lukas Broers  on 2024-09-30  [id 4809]

(in reply to Report 2 on 2024-02-19)

We thank the referee for this very detailed reply, and for their positive assessment, as well as recommending publication of our work.

A/
We thank the referee for drawing our attention to this reference. It describes a non-perturbative Boltzmann approach for describing the photoconductivity of extrinsic graphene for small driving frequencies. In particular, they consider the driving photon energy to be below the Fermi energy, and therefore interband transitions become negligible. The work is consistent with our work in the sense, that it is complementary to our approach as it describes a different regime of photoconductivity and treats it with a distinct method. We have included this reference in the introduction of our revision. We note that this reference is also included in the list of references provided by the other referee, who raises similar concerns. We have compared these references to the results of our work, and find that they do not take away from our results.

B/
We thank the referee for this suggestion. A direct and in-depth comparison to experimental results would be valuable. We understand that the proper way of doing so is in a separate collaborative study, which we find to be out of the scope of our presented manuscript.

C/
We thank the referee for this suggestion. Third-order susceptibilities and saturation effects capture very interesting optical responses that we might explore in detail in future work. In this manuscript, we set on focusing on the non-perturbative DC responses in the presence of a DC field. While it is true that in the edge-case of a vanishing DC field, it is possible to study the third-order susceptibilities of the AC photoconductivity, we find this to be a topic that warrants a separate analysis and publication. We note that the two references the referee provides are also included in the list of references provided by the other referee. We include these references and contextualize them in the introduction of the revised manuscript.

D/
Yes, the reduction is applied to all dissipation coefficients equally, such that it keeps the ratios between coefficients fixed. In the revision we are more precise in how we phrase this. Indeed, for very strong dissipation we expect the characteristic features of the photoconductivity to be gradually lost. At that point we expect the dynamics to be dominated by semi-classical carrier dynamics. We believe that this comparison would be a very interesting analysis for future work.

E/
We thank the referee for these remarks, as they improve the manuscript. As suggested, we have gone over the manuscript to make the language more consistent, and we have fixed the typo. We have also gone over the reference list and have corrected the improper use of lowercase characters.

---

## Round 2 · Referee Report · Anonymous (Referee 2) · 2024-10-5

Report

This is a single-blind review of once-revised SciPost Physics manuscript 2312.13217v2 titled "Non-linear photoconductivity of strongly driven graphene" by Lukas Broers and Ludwig Mathey. The revised manuscript is one page longer (from 27 to 28) and the bibliography has +13 entries (from 62 to 75), compared to the v1 submission; the number of figures and equations has not changed. The manuscript can be viewed in arxiv https://arxiv.org/abs/2312.13217v2.

The Authors have revised the manuscript but, apart from the increase in the bibliographic entries (requested by an easily guessable referee), the other changes were only textual improvements and minor elaborations. For that matter, a red-lined version would be more appropriate to locate these changes in the text, instead of having to read-and-compare the two manuscripts. Moreover, a number of the points I raised were off-loaded (read: "not sufficiently addressed") as possible/interesting future works/directions, and not for the improvement of the present manuscript. Specifically, concerning the points I raised in my first review:

A/ Apart from the mentioned qualitative comparison with existing literature on the topic, and any (short) textual discussions, what does "we find that they do not take away from our results" mean? Is there some quantitative comparison, even preliminary, to corroborate that claim? This is essential for the novelty of the paper. This comparison could be provided in a response letter or in a supplement (to the manuscript), i.e., it does not need to be "camera-ready".

B/ What I meant was to look in the literature for measured/experimental results that can be interpreted with this model, even qualitatively (e.g., does it predict the same trends?). Please note that one of the expectations for papers in SciPostPhys [ https://scipost.org/SciPostPhys/about#criteria ] is to "Provide a novel and synergetic link between different research areas", and such a comparison between theory and experiment would certainly comply with that criterion.

C/ Similarly to my comment B, a short/preliminary analysis on that matter (or at least something more than just "contextualization" in the text) would improve the manuscript's scope and positioning in the related state-of-the-art.

D/ Thank you for the clarification. Please also consider what I remark in my previous comments B and C.

Recommendation

Ask for minor revision

  • validity: -
  • significance: -
  • originality: -
  • clarity: -
  • formatting: -
  • grammar: -

Author:  Lukas Broers  on 2025-05-06  [id 5446]

(in reply to Report 1 on 2024-10-05)

We thank the referee for their reply.

A)
Let us clarify what we meant by the statement that the provided reference does not take away from our results. In their original report, the referee raised a question regarding the consistency of our results with those in the literature (this included a reference originally also provided by the first referee). The point we tried to make was that said reference considered transport in graphene in a regime, and under conditions, that are different from our work. The results do not take away from each other in the sense that they talk about different things. In particular, the provided reference considers a non-zero chemical potential and a driving frequency specifically below the chemical potential. That setup is not applicable to our study. That is why we felt justified in keeping our discussion short. Our numerical approach is in principle capable of simulating a non-zero chemical potential, and performing a study of this type would be possible, but this would amount to a different study of its own.

B)
We apologize for the misunderstanding. While we consider the parameters in our work to be realistic, we believe that the high driving field strengths and the low dissipation make a direct comparison with existing literature scarce. With regards to the characterization of graphene quality however, as mentioned in the original report by the referee, the undriven non-linear transport behavior is most relevant. Our results imply an I-V curve that changes from initial linear scaling to a power-law scaling for larger voltages. In particular, the current with respect to the bias voltage approaches $I \propto V^{\frac{3}{2}}$, which is consistent with the non-linear trend in experimentally measured I-V curves. Some examples can be found in these references:
https://doi.org/10.1016/j.apsusc.2018.04.099 - Fig. 6
https://doi.org/10.1109/ISOT.2009.5326118 - Fig. 7
https://doi.org/10.1016/j.jscs.2018.03.004 - Fig. 7
We have included a remark to this and some of these references in the revised manuscript.

C)
The referee was correct in their initial report that these quantities are related to our model in the limit of a vanishing DC field. Energy-dependent relaxation rates are currently not contained within our model. Nevertheless, third-order effects would emerge at comparatively low AC field strengths compared to the entire range of Fig. 2. They would however require a modified approach to evaluating the current, in order to separate the contributions from different effects. So while it is in principle within reach of our method, it is not clear what a short analysis on such matters would contain, as we would have to notably alter the methodology to approach such a comparison. In their original report, the referee stated we 'could consider' comparing our results to specific quantities found in non-linear optics. We still believe a proper comparison would justify its own study, and extending the scope of this manuscript in that direction would also not necessarily improve its accessibility. We are not trying to deflect the enquiry of the referee, but a serious comparison seems out of scope and also does not constitute a minor revision. This was the motivation behind our initial reply.

---

## Round 2 · Referee Report · Anonymous (Referee 3) · 2025-2-7

Strengths

(1) Application of the Lindblad-von Neumann master equation to the study of current response to electrical field in the presence of dissipation is a novel theoretical methodology.

Weaknesses

(1) It is not clear in the text how the authors separate the linear and nonlinear responses.
(2) Authors seem to lack some basic knowledge about graphene model they are treating in the paper.
(3) Definitions of conductivities are unclear and different from the concepts generally accepted by the physics community.

Report

The authors presented their theoretical and numerical work on the computation of the electrical conductivity of graphene driven by strong DC and AC electrical fields. By applying the Lindblad-von Neumann master equation describing the time-evolution of the density matrix, they tried to compute the current density and then the conductivity by means of a differential expression.

Here are some questions and comments:

(1) At the beginning of the “Methods” section, the authors mentioned “A similar model applies to the other Dirac point, with $\sigma_y \rightarrow \sigma_y$”, which is not correct. In this work they have considered the model near the K point of graphene, where the Hamiltonian can be simplified to a Dirac equation $v_F(k_x \sigma_x+ k_y\sigma_y)$. While for the other Dirac point at K’, the Dirac equation is a “time reversed” version of the one near K point, meaning that not only $\sigma_y \rightarrow \sigma_y$, but also $k_x \rightarrow -k_x$ and $k_y \rightarrow -k_y$. As a result, the Hamiltonian at K’ point is $vF(-k_x\sigma_ x+ k_y \sigma_y)$. Then it leads to a problem in Eq. (3) : the current operator from K’ point of graphene gets a minus sign: $j_{x,K'} = - e v_f \sigma_x$. The total current as a response to the external field is a sum of the current from both valleys. As a consequence, it raises a question: upon applying an external field, are the current responses at K and K’ points exactly the same? Furthermore, DC current is known to make the electron population polarized between the valleys. In the presence of DC current, the current response to the AC field is a priori expected to be different from both valleys.

(2) The definition of the differential photoconductivity shown in Eq.(7) is quite confusing, which is in fact a second derivative of the current to the electrical field. This is very different from what is commonly known for electrical conductivity. Can the authors justify the necessity of this definition, as well as its link to the usual definition of linear and nonlinear conductivities as response functions?

(3) Although the nonlinear DC response is talked about throughout the manuscript, it is not clear how the authors separate the linear and nonlinear parts in the current. In terms of the nonlinear response, to which order do the authors consider?

To summarize, the authors have presented interesting methods based on the Lindblad-von Neumann master equation. But the basis on which to perform all these calculations and analyses is not clearly stated and not entirely correct and solid, showing that the authors may lack some important understanding about the graphene model and the theory of electrical conductivity. Therefore, I would not recommend the publication of the manuscript in any journal before the authors could correctly treat the graphene model and the linear and nonlinear responses to electrical fields.

Requested changes

(1) The current contribution from both valleys should considered separately, unless explicit justification shows they are identical.
(2) The authors should consider proper definition of the linear and nonlinear electrical conductivity and perform their analyses on the correct basis.

Recommendation

Reject

  • validity: low
  • significance: low
  • originality: good
  • clarity: poor
  • formatting: good
  • grammar: excellent

Author:  Lukas Broers  on 2025-03-24  [id 5308]

(in reply to Report 2 on 2025-02-07)

We thank the referee for their effort in evaluating our manuscript, and appreciate the positive assessment of our numerical method. We would like to address the remarks made by the referee, because we believe that we can clarify these misconceptions.

1) The referee remarks on the difference between the Dirac points, and how it might affect the results. The referee is right to be cautious, since in some situations such as in graphene nanoribbons or past the linearization around the Dirac points, these considerations become important. In our case we consider extended graphene and linearly polarized driving parallel to the direct bias field. It is easy to see that the two Dirac points contribute equally to the total current, which resolves one of the main criticisms by the referee. Continuing the reasoning the referee puts forth, the time-dependence of the vector potential is reversed as well. The driving term is linearly polarized and hence unaffected. The direct bias term collects a minus sign, such that it now acts as if the electric field had been reversed, in turn the current density obtains an additional minus sign, which will cancel that of the definition of the current, which the referee had mentioned. The only difference that remains is the location of the currents, the current-density resolved in momentum space is symmetric under $k_x \rightarrow -k_x$. However to obtain the total current, we integrate over k-space, such that the total current ends up identical at both Dirac points.

This result should be unsurprising, because we could imagine changing the roles of the spatial directions by applying both fields, as well as determining the current, along the y-direction instead of the x-direction. If we then take the corresponding Hamiltonian and perform the transformation suggested by the referee to obtain the K’ point, this ends up being equivalent to what we do in the manuscript, just under the exchange of x and y. Of course changing the direction of the entire electric field in this setup does not change the conductivity, because the linearized Hamiltonian is rotationally symmetric.

We will gladly clarify the phrasing regarding the other Dirac point in the manuscript, if it is considered necessary.

2) The referee is correct that we are presenting a second derivative. This is the general way to express how an observable changes with respect to a system parameter. The definition of electrical conductivity as linear- and higher-order non-linear terms comes from the expansion coefficients of the Taylor-expansion of the derivative as we present it. We think the definition in terms of derivatives is justified by its generality. The correct way to think about this quantity and consequently linear and non-linear responses is to consider the general expansion in orders of the two applied fields: $\partial_{E_d}\partial_{E_0} j_x = \chi_{1,0} E_0 + \chi_{0,1} E_d + \chi_{1,1} E_0 E_d + \chi_{2,0}E_0^2 + \chi_{0,2}E_d^2 + \chi_{1,2} E_0 E_d^2 + \chi_{2,1} E_0^2 E_d +\chi_{2,2} E_0^2 E_d^2 + \dots$

Generality aside, the reason we believe our expression is well-chosen in this work, is that in our model it is numerically very naturally obtained. There is no explicit need to focus on the specific orders of non-linearity if we care about the behavior across very large regimes across these two electric fields.

3.) We hope our explanation above also clarifies this point, in which the referee asks how we separate linear and non-linear parts of the current. Whenever we write about linear response, we refer to the limit of $E_0\rightarrow 0$ in the sense that only the linear term in the Taylor-Expansion given above remains, as is consisted with basic linear response theory. We have pointed this out in the manuscript. Regarding the question of what orders of non-linearity we consider, we refer to our reply to the previous remark. In analysing the differential conductivity we technically consider all orders, but we do not separate them into individual susceptibilities, because are interested in the general behavior rather than some specific non-linear order.

Anonymous on 2025-03-28  [id 5323]

(in reply to Lukas Broers on 2025-03-24 [id 5308])

Thank the authors for their efforts of clarifying my questions. Here are my comments: 1) the graphene model near K and K' valleys are time-reversed counter parts, but the electric field is not time reverse. The authors should justify the following two Hamlitonians give the same current: $H_K(\mathbf r ,t) = \hbar v_F \mathbf k \cdot \mathbf \sigma + V(t)$ and $H_{K'}(\mathbf r ,t) = - \hbar v_F \mathbf k \cdot \mathbf \sigma^* + V(t)$ . $V(t)$ is the electric potential. Note that time reversal should not be applied to the electric field $V(t)$. Both valleys are under the same field, and the outcome is the sum of current from both valleys. Or I would suggest limiting the discussion to one specific valley, say the $K$-valley and remove the valley degeneracy, if this makes life easier.

2) From the formula given in the reply from the authors, are you considering linear and nonlinear effects on a second derivatitve of the current with respect to the electric field amplitude? However this makes the "conductivity" talked about in this paper more confusing. Conductivity tensors are functional derivatives of current with respect electric field, instead of simple derivatives with respect to the amplitude of the field. To avoid the functional derivative one can go to the Fourier space. Eq.(4) is equivalent to evaluating the zero-frequency compenent of the current, i.e. the DC current. In terms of second order nonlinear effect, this is related to shift/injection current. I would expect the this second order conductivity to be also sensitive to the AC frequency. I would suggest the authors to comment more in the manuscript about the how the nonlinear conductivity varies with the AC frequency and why the specific AC frequency was chosen.

Anonymous on 2025-05-06  [id 5445]

(in reply to Anonymous Comment on 2025-03-28 [id 5323])

We thank the referee for their reply.

1.)
We see how our notation would benefit from some more detail. Our definitions of the current operator and the Hamiltonian are with regards to the K-point, but should be expressed in a more general way (as pointed out by the referee in their initial report). We improve this in the revised manuscript. Regarding the two Dirac points, we believe the argument from symmetry still holds. The notation of V(t) is somewhat ambiguous, so let us write:

$H_{K }(\mathbf{k}, t) = \hbar k_x \sigma_x + \hbar k_y \sigma_y $
$H_{K'}(\mathbf{k}, t) = -\hbar k_x \sigma_x + \hbar k_y \sigma_y$

Introducing the vector potential via Peierl's substitution / minimal coupling, we obtain
$H_{K }(\mathbf{k}, t) = \hbar (k_x - A_x) \sigma_x + \hbar k_y \sigma_y $
$H_{K'}(\mathbf{k}, t) = -\hbar (k_x - A_x) \sigma_x + \hbar k_y \sigma_y$

The Hamiltonian at the K'-point can be reexpressed as
$H_{K'}((-k_x,k_y), t) = \hbar (k_x + A_x) \sigma_x + \hbar k_y \sigma_y$

Near the K'-point, we now find a vector potential that is effectively reversed compared to the K-point. This leads to a current density that would produce a current opposite to that of the $K$-point, if it weren't for the additional minus sign in the definition of the current operator. This leads to the same current density as at the K-point, however mirrored due to $k_x\rightarrow -k_x$. Since we consider the integrated current over $k$-space, the fact that the current density is mirrored does not affect the total result. As a side note, if this argument were not to hold, then even if $A_x$ only consisted of a small DC field, the currents would cancel and there would be no conductance along the $x$ direction (while in contrast there would be conductance along the $y$ direction if $A_x=0$ and $A_y$ corresponded to a small DC field). Furthermore, the driving term could at most lead to an out-of-phase response at the two Dirac points, but since we are integrating over one driving cycle in the steady state to obtain the total current this does also not affect the results. We make this point more clear in the revision.

2.)
Yes, we are considering the conductivity in terms of the derivative with respect to the electric field amplitudes. The referee is of course correct, that in general this would be a functional derivative, but also that this can be avoided in Fourier space. In our Hamiltonian, the electric field has two components, one at zero-frequency and one at the driving frequency. Their amplitudes are the ones with respect to which we take the derivatives in the definition of the conductivity. So we agree with the referee here. We concede that this point is left implicit in our manuscript. In the revision we have clarified this.

Keeping every parameter fixed and tuning the driving frequency would display a frequency-dependence in the conductivity. This can however be better understood as rescaling the conductivity as a function of $E_0$ and $E_d$. If we rescale the driving frequency as $\omega_d \rightarrow \alpha \omega_d$, then we can recover the original dynamics by rescaling $E_d \rightarrow \alpha^2 E_d$, $E_0 \rightarrow \alpha^2 E_0$, and $\gamma\rightarrow\alpha\gamma$, as well as $k\rightarrow \alpha k$ and $t \rightarrow t/ \alpha$. Therefore, when changing the driving frequency we would find the same qualitative behavior as in Fig. 2, with rescaled axes and dissipation coefficients.

The driving frequency in our work is inspired by previous works [PhysRevResearch.2.043408] and was originally motivated from experiment. We have reduced the frequency slightly from that original value due to reasons that are related to the previous point. With a larger frequency, we also need larger field strengths to map out Fig. 2, which at some point approach values that we deemed unfeasible. Also in that case, the current density occupies a much larger region in momentum space, which makes the numerical simulation more demanding. At the same time there are reasons why reducing the frequency is undesirable. We want to stay close to the original experimental setup, and also want to keep the dissipation coefficients at realistically achievable values. The dissipation coefficients however, have to be reduced when reducing the driving frequency in order to still resolve the features in Fig. 2. The value we have chosen in the manuscript is a balance between these considerations.

---

## Round 2 · List of Changes

We have included references and discussions of the literature provided by the two referees, to better contextualize our work. We have included a clarifying remark regarding the formation of the current density patterns. We have further included remarks regarding the dissipation scaling, and we have reworked the text to increase the consistency of our terminology as suggested by one referee. We have fixed the typo and the reference list.

---

## Round 3 · Referee Report · Anonymous (Referee 3) · 2025-5-6

Report

In this new version, the authors have clarified issues concerning previous questions.
The methodology presented in this work is interesting and novel.
I think the manuscript meets the requirement for publication in its current form.

Recommendation

Publish (meets expectations and criteria for this Journal)

---

## Round 3 · Referee Report · Anonymous (Referee 2) · 2025-5-20

Report

No further comments. The Authors have addressed the points I raised on my previous review in a satisfactory manner.

Recommendation

Publish (meets expectations and criteria for this Journal)

---

## Round 3 · List of Changes

• Changed the notation of Eq. 1 and Eq. 3 to include the difference between both Dirac points
  • Included clarifying remarks on the current contributions of the two Dirac points
  • Included clarifying remark on the differential conductivities
  • Included remark on the agreement of the non-linear conductivity with experimental measurements

---

## Editorial Decision

published